# Singlet fission as a polarized spin generator for dynamic nuclear polarization

Yusuke Kawashima[1,12], Tomoyuki Hamachi[1,12], Akio Yamauchi[1], Koki Nishimura[1], Yuma Nakashima[1], Saiya Fujiwara [1], Nobuo Kimizuka [1,2], Tomohiro Ryu[3], Tetsu Tamura[3], Masaki Saigo[3], Ken Onda [3], Shunsuke Sato[4], Yasuhiro Kobori [5,6], Kenichiro Tateishi[7,8], Tomohiro Uesaka[7,8], Go Watanabe [4,9] ✉, Kiyoshi Miyata [3] ✉ & Nobuhiro Yanai [1,2,10,11] ✉

Singlet fission (SF), converting a singlet excited state into a spin-correlated triplet-pair state, is an effective way to generate a spin quintet state in organic materials. Although its application to photovoltaics as an exciton multiplier has been extensively studied, the use of its unique spin degree of freedom has been largely unexplored. Here, we demonstrate that the spin polarization of the quintet multiexcitons generated by SF improves the sensitivity of magnetic resonance of water molecules through dynamic nuclear polarization (DNP). We form supramolecular assemblies of a few pentacene chromophores and use SF-born quintet spins to achieve DNP of water-glycerol, the most basic biological matrix, as evidenced by the dependence of nuclear polarization enhancement on magnetic field and microwave power. Our demonstration opens a use of SF as a polarized spin generator in bio-quantum technology.

Photo-excited states of organic assemblies have brought a number of unique opportunities to optoelectronics, taking advantage of the dual nature of singlet and triplet molecular excitons[1]. In particular, singlet fission (SF)[2–12], which generates two triplet excitons from one singlet exciton, shows unique functions in terms of electron and spin degrees of freedom. SF is a multiexciton generation process that can potentially surpass the theoretical limit of a single-junction solar cell if the split excitons are harvested as free electrons and holes[5]. Its unique electron degree of freedom has attracted much attention and has been studied intensively for decades.

The basic SF process is as follows: the singlet exciton $S_1$ undergoes a spin-allowed ultrafast transition to a triplet-pair state with overall-singlet multiplicity $^1(TT)$, followed by an intersystem crossing (ISC) to the highest spin multiplicity state, a quintet triplet-pair $^5(TT)$. When the molecular assembly is larger than two molecules, the triplet-pair states may dissociate into two free triplets[9,13–16]. Note that the multiexcitonic nature offers a unique opportunity to construct quintet multiplicity owing to the presence of four half-filled orbitals. SF provides the effective method to create spin-polarized quintet states in organic systems without using heavy metals. However, how to use this unique quintet state has not been fully demonstrated.

We explore the unique spin degree of freedom of SF for quantum technologies[17,18]. Among the five quintet spin sublevels, it has been reported that certain sublevels can be preferentially populated[13–16]. According to the *JDE* model, the $^5(TT)_0$ quintet state of chromophore dimers can be generated as a nearly pure quantum state by making the

[1]Department of Applied Chemistry, Graduate School of Engineering, 744 Moto-oka, Nishi-ku, Fukuoka 819-0395, Japan. [2]Center for Molecular Systems (CMS), 744 Moto-oka, Nishi-ku, Fukuoka 819-0395, Japan. [3]Department of Chemistry, Graduate School of Science, Kyushu University, 744 Moto-oka, Nishi-ku, Fukuoka 819-0395, Japan. [4]Department of Physics, School of Science, Kitasato University, 1-15-1 Kitazato, Minami-ku, Sagamihara, Kanagawa 252-0373, Japan. [5]Molecular Photoscience Research Center, 1-1, Rokkodai-cho, Nada-ku, Kobe 657-8501, Japan. [6]Department of Chemistry, Graduate School of Science, Kobe University, 1-1, Rokkodai-cho, Nada-ku, Kobe 657-8501, Japan. [7]Cluster for Pioneering Research, RIKEN, 2-1 Hirosawa, Wako, Saitama 351-0198, Japan. [8]RIKEN Nishina Center for Accelerator-Based Science, 2-1 Hirosawa, Wako, Saitama 351-0198, Japan. [9]Kanagawa Institute of Industrial Science and Technology (KISTEC), 705-1 Shimoimaizumi, Ebina, Kanagawa 243-0435, Japan. [10]PRESTO, JST, Honcho 4-1-8, Kawaguchi, Saitama 332-0012, Japan. [11]FOREST, JST, Honcho 4-1-8, Kawaguchi, Saitama 332-0012, Japan. [12]These authors contributed equally: Yusuke Kawashima, Tomoyuki Hamachi. ✉e-mail: go0325@kitasato-u.ac.jp; kmiyata@chem.kyushu-univ.jp; yanai@mail.cstm.kyushu-u.ac.jp

exchange interaction between two chromophores sufficiently large and by making the principal axes of the two chromophores parallel to each other and to the Zeeman field[19]. This model has explained well the experimental results of oriented crystalline samples[20,21]. To date, the spin aspect of SF has only been used to explain the microscopic mechanisms of SF. Because organic spin materials have advantages with their extremely small size, down to nanometers, and excellent bio-compatibility, it is worthwhile to research applications of the unique quintet state in quantum information science (QIS) and quantum biotechnologies[17–19].

Dynamic nuclear polarization (DNP) of biomolecules is one of the fields where polarized electron spins can play a pivotal role[22–29]. Nuclear magnetic resonance (NMR) and magnetic resonance imaging (MRI) are indispensable analytical techniques in modern life science and medicine, but their critically low sensitivity limits their applications. Many clinical trials of MRI diagnosis of cancer have been conducted by transferring the polarization of radical electron spins, which are in thermal equilibrium at cryogenic temperatures near 1 K, to the nuclear spins of bioprobes, and then dissolving them and administering them to the human body[30]. However, the equipment is inevitably expensive and complicated because it requires cryogenic temperatures. Thus there is a strong need to develop DNP using polarized electron spins generated at higher temperatures. The triplet excited state generated by spin-preferential ISC from a photoexcited singlet has been used as a polarized spin source[31], but the polarization ratio of the triplet is usually far less than 100%[32]. However, SF has the potential to offer the ultimate polarization source because it can preferentially populate the $^5(TT)_0$ spin sublevel by appropriately controlling the structure and orientation of chromophore dimers[19].

This study demonstrates the application of a SF-born polarized quintet state in DNP of water-based glass. Various small biomolecules and proteins can be hyperpolarized by dispersing them in an amorphous water-glycerol glass[22]. The key to successful DNP in a biooriented water-glycerol environment is to regulate the balance between aggregation and dispersion of SF molecular assembly; more than two molecules are needed for SF, but aggregation that is too large hampers the polarization transfer from the SF-generated electron spins to the nuclear spins. In essence, creating robust dimer aggregates would be promising to achieve both efficient SF and polarization transfer to the water-glycerol and eventually to dispersed biomolecules. We focus on a pentacene derivative, the most representative chromophore exhibiting SF[8]. We succeed in constructing discrete assemblies of pentacene moieties in water-glycerol using two distinct strategies: supramolecular assembly of an amphiphilic pentacene derivative and complexation with cyclodextrin (CD)[33]. The combination of ultrafast pump-probe transient absorption spectroscopy (TAS) and time-resolved electron spin resonance (ESR) measurements reveals that either pentacene assembly undergoes SF and generates electron spin polarization. We succeed in DNP of water-glycerol by transferring the polarization from quintet electron spins to nuclear spins upon microwave irradiation to satisfy the Hartmann–Hahn condition (Fig. 1B, C). In addition, we show that the magnetic field dependence of nuclear polarization enhancement matches well with the ESR line shape and quintets with higher Rabi frequencies can cause DNP at lower microwave intensities than a conventional triplet[17], confirming the DNP based on the polarized quintet spins.

## Results and discussion

To construct supramolecular assemblies showing SF, we used our previously developed amphiphilic sodium 4,4′-(pentacene-6,13-diyl) dibenzoate (NaPDBA, Fig. 1D): a hydrophobic pentacene modified with hydrophilic carboxyl groups. We molecularly dispersed 1-mM NaPDBA in methanol and obtained an absorption peak at 593.5 nm. In water-glycerol, the absorption peak was clearly red-shifted to 604 nm, suggesting the formation of supramolecular assemblies (Fig. 1E). We

adjusted the mixing ratio of water and glycerol and used a 1:1 mixture by volume because it maintains the glassy state in the DNP-relevant condition, i.e., low temperature under laser irradiation. Molecular dynamics (MD) simulation was performed for an initial structure of 20 molecules of NaPDBA in close proximity to each other in water-glycerol. The simulation showed that the structure split into multiple dimers and a few monomers (Fig. 2A, B). Supplementary Fig. 1 shows the time variation of the distance between the centers of the mass ($d_{COM}$) of pentacene and the angle ($\theta$) between $d_{COM}$ and the nearest distance between the pentacene units ($d_{min}$) obtained from the MD simulations of NaPDBA in water-glycerol at 300 K. The distance between pentacene units is relatively far with various orientations, which probably gave a slightly red-shifted and sharp absorption peaks. The formation of small dimeric structures agrees well with the fact that dynamic light scattering (DLS) measurements of NaPDBA in water-glycerol did not show any significant scattering intensity derived from large aggregates. However, DLS measurements showed that NaPDBA forms larger structures in pure water without glycerol (Supplementary Fig. 2). MD simulations also confirmed the formation of stable NaPDBA multimers in water (Supplementary Fig. 3). The addition of glycerol may have weakened the hydrophobic interactions between NaPDBA monomers and prevented the formation of larger aggregates, leading to the formation of NaPDBA dimers[34].

To systematically change the assembly structure and excitonic interaction between the pentacene moieties while using the same NaPDBA molecule, we added β-cyclodextrin (βCD) and γ-cyclodextrin (γCD) to NaPDBA in water-glycerol. βCD and γCD are cyclic oligosaccharides with seven and eight glucose subunits, respectively, and can host various hydrophobic compounds within their hydrophobic interiors. The addition of γCD did not change the absorption spectrum of NaPDBA at room temperature with an absorption peak at 604 nm (Supplementary Fig. 4). Notably, the absorption peak of NaPDBA was further red-shifted to 612 nm and broadened by cooling to 143 K after inclusion was allowed to fully progress by letting the solution stand at 243 K in the presence of γCD (Fig. 1E and Supplementary Fig. 4). This suggests that the excitonic interaction between pentacenes was enhanced by the formation of the inclusion complex between NaPDBA and γCD at low temperature. NMR studies of NaPDBA and γCD suggested that two molecules of γCD encapsulate a NaPDBA dimer, which is reasonable considering the large inner diameter of γCD and the strong excitonic interaction between pentacene moieties in the NaPDBA-γCD inclusion complex (see Supplementary Information for details, Supplementary Figs. 4–15). In the MD simulation, the 2:2 inclusion complex of NaPDBA-γCD was stable in water-glycerol at 243 K (Fig. 2C, D). The MD simulation results showed that both $d_{COM}$ and $\theta$ fluctuate less in NaPDBA-γCD than in NaPDBA alone, which is reasonable since NaPDBA dimers are encapsulated in γCD (Supplementary Fig. 1). In NaPDBA-γCD, the pentacene units associate at a closer distance, causing orbital overlap between the two chromophores. In the presence of orbital overlaps, we cannot simply classify J- or H-aggregate assuming point-dipole approximation; the excitonic coupling depends sensitively on the relative geometry between the chromophores and molecular orbitals. It has been reported that a few Å displacements affect the sign of the inter-chromophore interaction, demonstrating both J-aggregate-like and H-aggregate-like spectral changes at face-to-face geometry[35]. According to our MD simulation, the shift along the long-axis of the pentacene backbones varies by a few Å. This variation would result in the simultaneous presence of J-aggregate-like and H-aggregate-like dimers in the system, resulting in the broad absorption spectra of NaPDBA-γCD. Note that the inclusion complexes of NaPDBA-γCD were unstable at room temperature in water-glycerol (Supplementary Fig. 12), which is consistent with the experimental results for complexation only by cooling (Supplementary Fig. 4). The analysis of the potential of mean force (PMF) of the complexes also supported this stability difference quantitatively: the

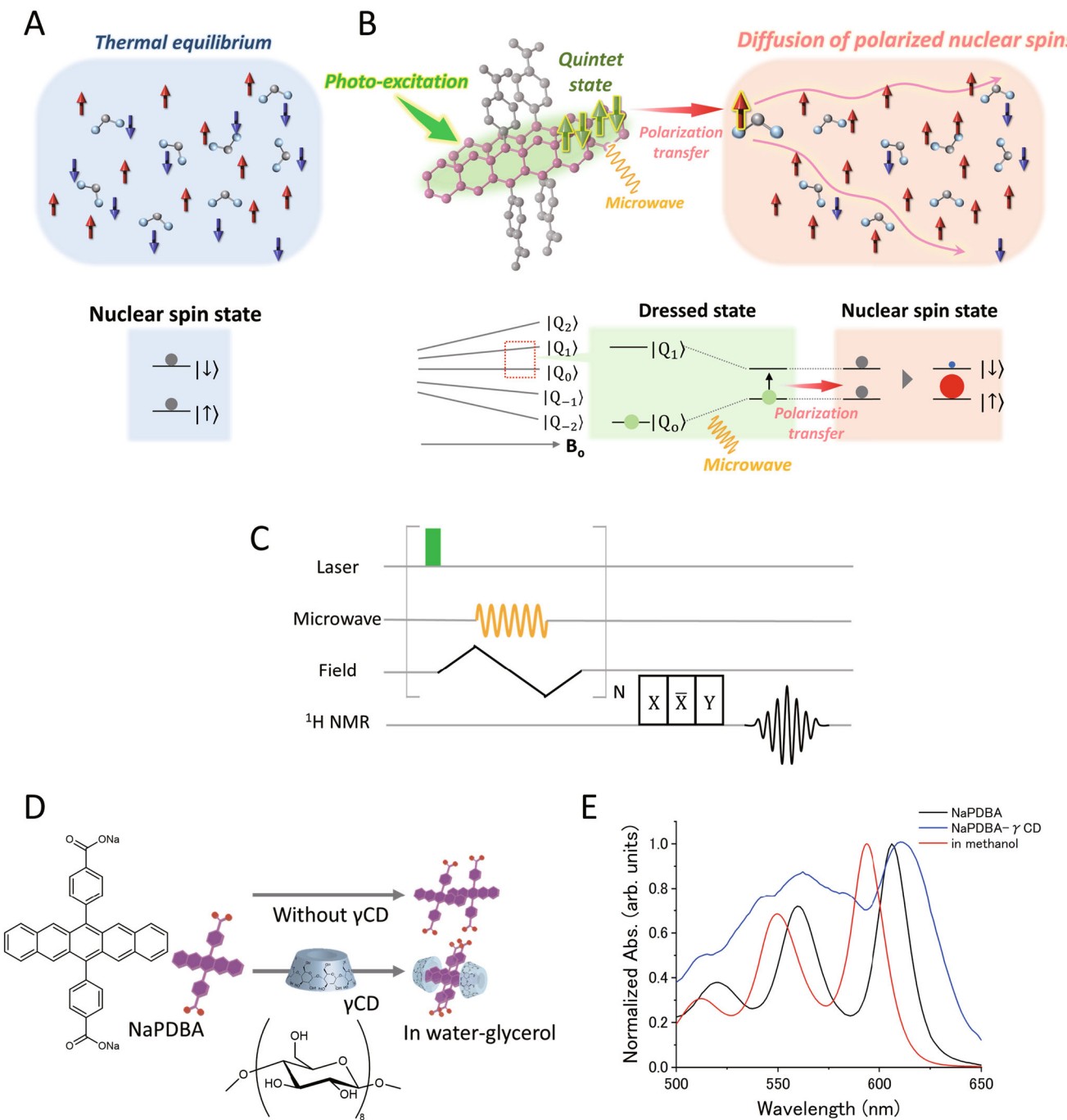

**Fig. 1 | Schematic illustration of DNP using SF-born quintet electron polarization.** **A** Nuclear spins in the thermal equilibrium state. The red and blue arrows indicate α spin state and β spin state, respectively, and the gray circles indicate the populations of each spin state. **B** Polarization transfer from electron spins in the quintet state ($|Q_0\rangle$) state, green arrows) generated by photo-induced SF to nuclear spins and the subsequent diffusion of hyperpolarized nuclear spins. The green circles indicate the populations of polarized quintet state. DNP increases the α spin population (red circle) and decrease the β spin population (blue circle), resulting in the hyperpolarized nuclear spin state (red square). **C** Pulse sequence of quintet/triplet-DNP. **D** Molecular structures of NaPDBA and γ-cyclodextrin (γCD) and supramolecular assembly of only NaPDBA and the NaPDBA-γCD inclusion complex. **E** Absorption spectra of NaPDBA in water-glycerol at 143 K (black), NaPDBA-γCD in water-glycerol (1:1) at 143 K (blue), and NaPDBA in methanol at room temperature (red). The concentrations of NaPDBA and γCD were 1 and 5 mM, respectively.

higher the PMF value of the complex, the more stable the complex. The PMFs for pulling away one γCD molecule of the 2:2 inclusion complex of NaPDBA-γCD in water-glycerol until it unfolded the NaPDBA dimer were $7.9 \pm 2.1$ and $21.1 \pm 2.4$ kJ/mol at 300 K and 243 K, respectively (Supplementary Fig. 15). Therefore, it shows that the NaPDBA-γCD inclusion complex at 243 K was more stable than that at 300 K. Since the thermal energy of the complex consisting of 4 molecules at 300 K

can be estimated about 10 kJ/mol, the complex of NaPDBA-γCD in water-glycerol at 300 K should not be energetically stable. The inclusion of the NaPDBA monomer by two βCD molecules was suggested by the absorption spectra, NMR measurements, and MD simulations (see Supplementary Information for details, Supplementary Figs. 4–15). This 1:2 inclusion complex of NaPDBA-βCD without SF was used for comparison.

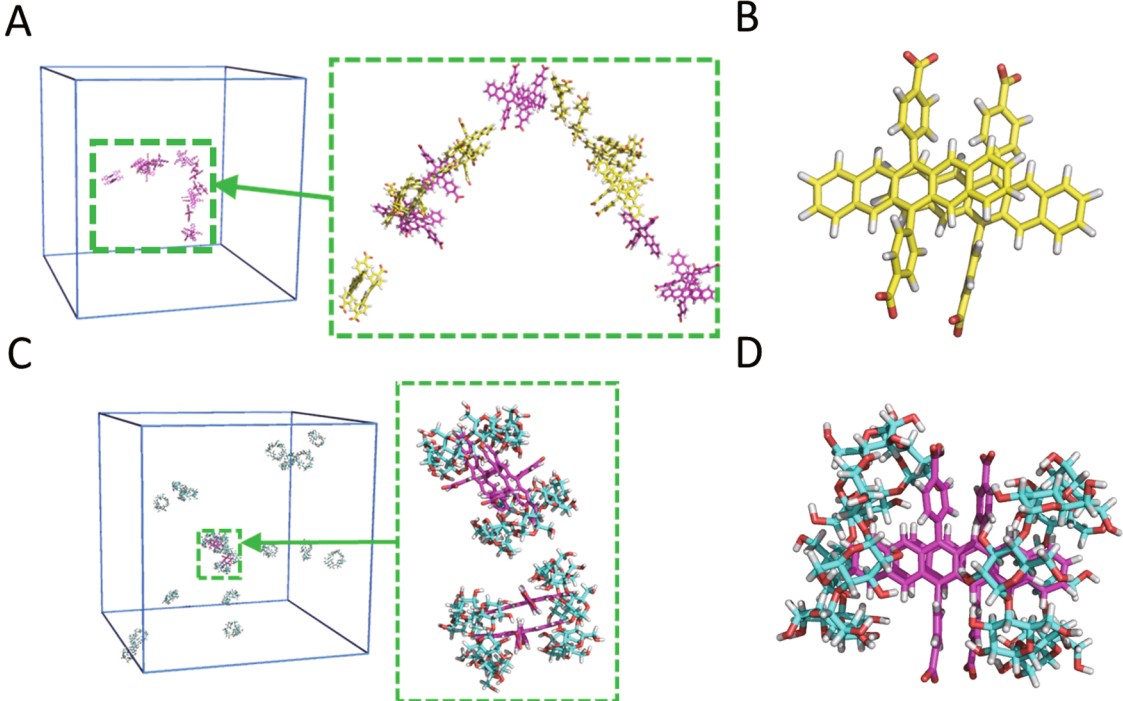

**Fig. 2 | MD simulation of the supramolecular assemblies. A, B** MD simulation snapshots of NaPDBA ([NaPDBA] = 1 mM) in water-glycerol (1:1) at 300 K. Parallel oriented dimers are shown in yellow. **C, D** MD simulation snapshots of NaPDBA and γCD ([NaPDBA] = 1 mM, [γCD] = 5 mM) in water-glycerol (1:1) at 243 K.

## Evidence of SF

To investigate the SF properties of these systems, we conducted femtosecond and nanosecond pump-probe TAS (fs- and ns-TAS) measurements in water-glycerol glass at 143 K, which is relevant to ESR and DNP measurements. Prior to the TAS measurements, the water-glycerol solutions of NaPDBA and NaPDBA-βCD were quenched and vitrified in liquid nitrogen after complexation at 243 K. The excitation wavelength was set at the lowest absorption edge (600–635 nm) to minimize the excess photoexcitation energy. Figure 3 shows a thorough comparison of fs-TAS measurements of the systems and the corresponding results of global analyses. The bare NaPDBA aggregates showed broad transient absorption with a peak at 430–450 nm just after photoexcitation (Fig. 3A–C and Supplementary Fig. 16), which can be assigned to the transition from the $S_1$ excited state ($S_1$–$S_n$ transition). For pentacene-based systems, the presence of SF can be estimated from the ultrafast growth of transient absorption around 510–520 nm typically assigned to the $T_1$–$T_n$ transition of a pentacene skeleton[8]. As the peak around 450–460 nm decreased in several picoseconds, the TA around 510–520 nm became dominant. This allowed us to confirm the ultrafast generation of $T_1$ through SF of NaPDBA aggregates. To quantify the ultrafast SF process, we globally analyzed the observed fs-TAS assuming a sequential model with two components (Fig. 3C–E). The first component was converted to the second component with a time constant of $2.65 \pm 0.01$ ps, followed by negligible decay (>1 ns) in the current time window. Note that the first and second components of the evolution-associated spectra (EAS) can reasonably be assigned from their shapes to the spectra from $S_1$ and $T_1$, respectively. Because the transition timescale is much quicker than that of typical ISC, we concluded that the transition of the first step of SF, $S_1 \rightarrow {}^1(TT)$, in the NaPDBA aggregates occurs with a time constant of 2.65 ps. We also observed an increase in $T_1$–$T_n$ absorption for ~7 ns in ns-TAS, which was assigned to the ISC of monomolecularly dispersed NaPDBA (Supplementary Fig. 17). This implies that the system consisted of a mixture of monomeric and dimeric NaPDBA, consistent with the MD simulation.

For the NaPDBA-γCD complex, the TAS around 510–520 nm emerged rapidly after photoexcitation concurrently with the broad absorption around 450–500 nm (Fig. 3F–H). This indicated prompt generation of the excited states containing ${}^1(TT)$ states character owing to the stronger electronic coupling between the chromophores. The spectral shape of the initial TA was also different from that observed for the bare NaPDBA aggregates. Note that a significant red-shift of the TA peaks in the ps-time range was also observed, which cannot be explained by simply assuming the inhomogeneous broadening owing to different conformers of the NaPDBA-γCD complex. These observations can be explained by the model of either (1) initial generation of the $S_1$-TT mixed adiabatic electronic states as suggested in the chromophores with strong electronic interaction[36,37], or (2) simultaneous detection of coherent and incoherent SF[38], which could result from the dynamic fluctuation and inhomogeneity of the system.

Figure 3H–J show the results of the global fitting of the TAS assuming a three-component sequential model. Although the third EAS resembles the spectrum of ${}^1(TT)$ observed in the bare NaPDBA aggregates, the first and second EAS components show different spectral shapes from the EAS in the bare NaPDBA aggregates. Given that the complexation with γCD results in the tightly packed dimer, we conclude as follows. Initial transient absorption spectrum (EAS1) contains spectroscopic characters of both $S_1$ and TT state because of strong electronic coupling between the adjacent pentacene moieties leading to the mixed adiabatic state of $S_1$ and ${}^1(TT)$ or rapid SF within the time resolution of our system (~100 fs). Note that we denote the initial component as "[$S_1$-TT]" to emphasize the indistinguishably mixed character of the state indicated by the spectral shape. Consequently, the transition to the hot ${}^1(TT)$ state ([$S_1$-TT] → TT\*) occurs in $0.79 \pm 0.01$ ps via SF. Because the timescale is likely quicker than the reorganization of the relative geometry of the paired chromophores (e.g., excimer formation), further spectral change with a time constant of $96 \pm 1$ ps occurs (denoted as TT\* → TT$^{rel}$ in Fig. 3I, J). To distinguish the whole mechanisms of the SF in the NaPDBA-γCD system, more sophisticated spectroscopy, such as coherent two-dimensional

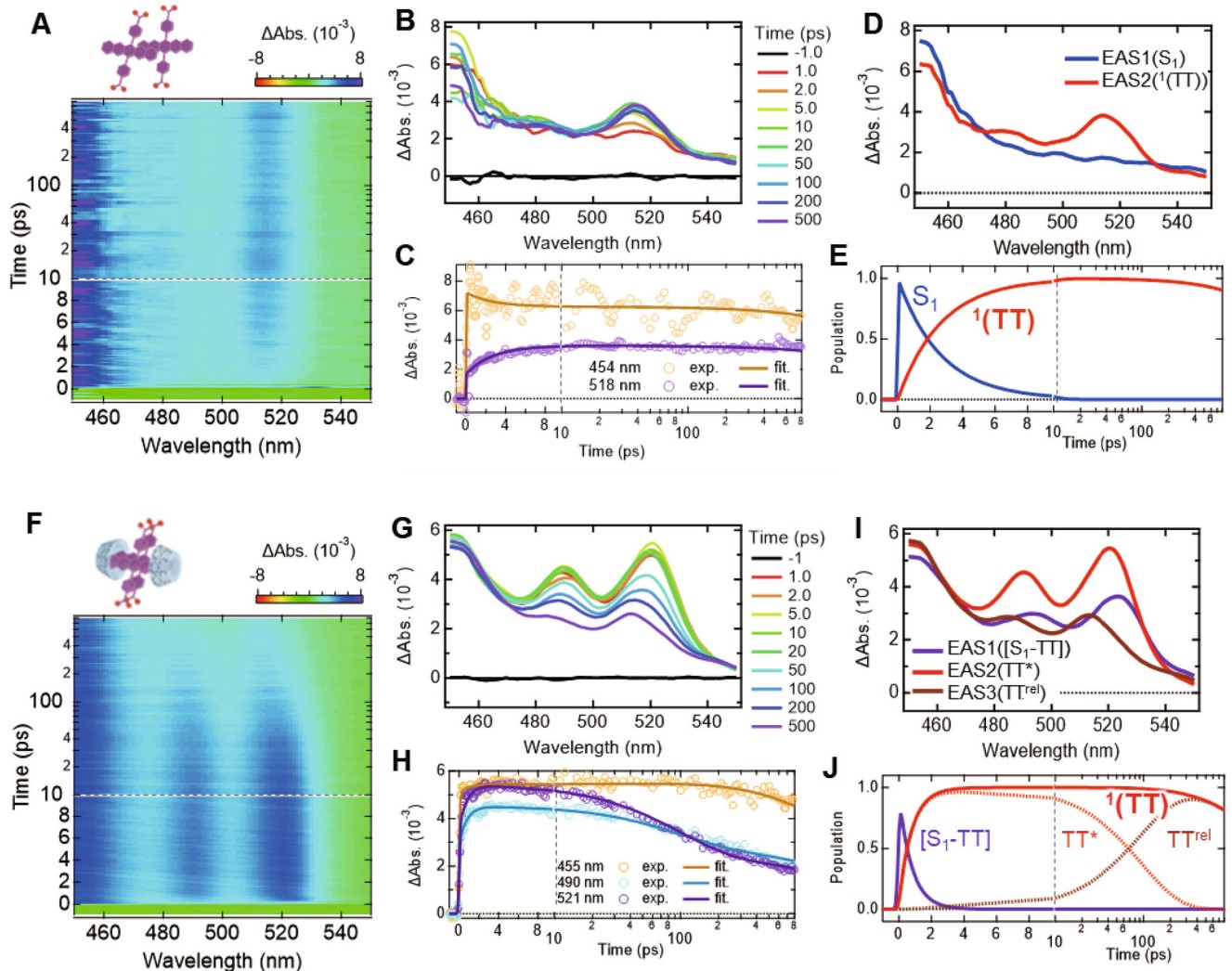

**Fig. 3 | fs-transient absorption spectroscopy (TAS) measurements of the supramolecular assemblies.** Overview of fs-TAS analysis of **A**–**E** NaPDBA and **F**–**J** NaPDBA-γCD in water-glycerol (1:1) at 143 K ([NaPDBA] = 1 mM, [γCD] = 5 mM). **A**, **F** Pseudo-2D plots of experimentally observed fs-TAS (excitation: 635 nm for NaPDBA and 600 nm for NaPDBA-γCD), **B**, **G** spectral evolution of the TAS, and **C**, **H** temporal change of transient absorption at selected wavelengths and fitting curves from global analysis. **D**, **I** Evolution-associated spectra (ESA) and **E**, **J** corresponding concentration kinetics obtained from global analysis based on sequential models. EAS1, EAS2, and EAS3 indicate the first, second and third components of EAS, respectively.

electronic spectroscopy, will be needed[39], which is beyond our focus in this manuscript. Here we emphasize that the TAS successfully confirmed the generation of ¹(TT) accelerated by more than a factor of three compared to the bare NaPDBA system.

In stark contrast, negligible SF was observed in the βCD complex (Supplementary Fig. 19), consistent with the monomolecularly dispersed picture in which the inter-chromophore interactions are significantly weakened by the complexation of a single NaPDBA molecule with two βCD molecules (Supplementary Fig. 11). We also observed the behavior in NaPDBA systems in water at room temperature (Supplementary Fig. 18), confirming that the dynamics is caused by complexation with cyclodextrins.

Overall, we concluded that sub-ns $T_1$ generation due to SF occurs in the bare NaPDBA dimers and NaPDBA-γCD complex, while the NaPDBA-βCD complex does not show significant SF. Note that ns-TAS measurements of the NaPDBA and NaPDBA-βCD systems also showed slower $T_1$ generation due to ISC from $S_1$ to $T_1$ (Supplementary Fig. 17). Additionally, fs-TAS and ns-TAS showed subtle excitation wavelength dependences (details are in Supplementary Fig. 20). This suggested that both aggregated and isolated NaPDBA coexisted in the water-glycerol glass systems, and both SF-derived and ISC-derived triplets

were generated with light irradiation. It would be possible to selectively make dimers or multimers by introducing functional groups to further control inter-chromophore interactions or by covalently connecting chromophore units.

## Generation of quintet multiexcitons

Having concrete evidence of SF, we performed time-resolved ESR measurements to evaluate the transient electron spin polarization of these supramolecular assemblies in water-glycerol glass. The samples were prepared in the same way as for the TAS measurements, except that glass capillaries were used. As expected from the SF found in the NaPDBA-only assemblies and the NaPDBA-γCD complex, a signal derived from the quintet was observed at a position where the peak width was about one-third that of the triplet (Fig. 4)[13–16]. These ESR spectra could be fitted as a quintet-triplet superposition. The spectra were simulated using a geometric fluctuation model of the quintet multiexcitons between two strongly coupled TT conformations ($TT_A$ and $TT_B$) with similar orientations with different $J$-couplings following previous reports[16,40]. We assumed the conformations of the $TT_A$ and $TT_B$ states to be parallel and slightly off parallel, respectively, for the vibronic effect of the $J$-coupling in the dimers. Interestingly, the

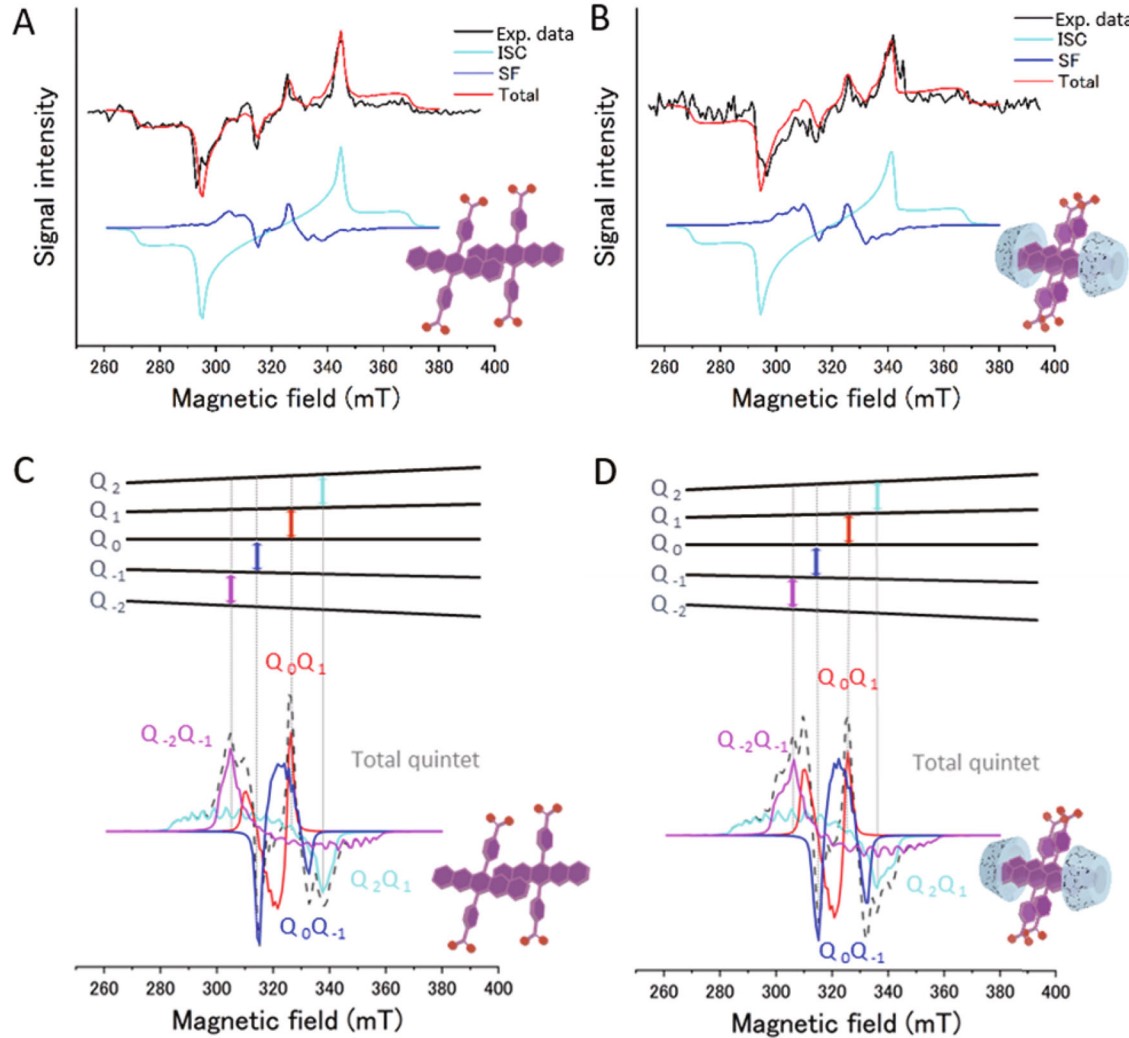

**Fig. 4 | Time-resolved ESR measurements of the supramolecular assemblies.**
Time-resolved ESR spectra of **A** NaPDBA and **B** NaPDBA-γCD in water-glycerol (1:1) at 143 K ([NaPDBA] = 1 mM, [γCD] = 5 mM) just after photoexcitation at 527 nm and simulated spectra of **C** NaPDBA and **D** for NaPDBA-γCD, attributing transitions between each energy level of the quintet in the ESR spectra. The fitting parameters of the ISC-born triplet and SF-born quintet are summarized in Supplementary Tables 1 and 2, respectively.

pentacene moieties were found to be oriented parallel to each other to form the $^5(TT)$ dimer, which agrees well with the MD simulations. The fitting parameters are summarized in Supplementary Tables 1 and 2. The $|^5(TT)_2\rangle$, $|^5(TT)_0\rangle$, and $|^5(TT)_{-2}\rangle$ sublevels were found to be preferentially populated with the states $S_1$ and $^1(TT)$ in both systems. The largest ESR signal was observed when the magnetic field ($B_0$) is parallel to the pentacene backbone (Supplementary Fig. 21). In this orientation, other states such as $^5(TT)_2$ and $^5(TT)_{-2}$ are also populated, however, ESR transitions involving $^5(TT)_2$ and $^5(TT)_{-2}$ have almost no contribution to this signal. To demonstrate this point, we computed ESR spectrum of $^5TT$ state obtained by the powder pattern calculation with considering the spin populations in all the sublevels $^5(TT)_{+2}$, $^5(TT)_{+1}$, $^5(TT)_0$, $^5(TT)_{-1}$ and $^5(TT)_{-2}$ (Supplementary Fig. 22A) and the powder pattern with only by $^5(TT)_0 \rightarrow {}^5(TT)_{+1}$ and by $^5(TT)_0 \rightarrow {}^5(TT)_{-1}$ contributions (Supplementary Fig. 22B). At the field strengths represented by "Z" and "X, Y" in Supplementary Fig. 22A, the ESR transition intensities (transverse magnetizations) by the quintet states are dominated by the resonances from the $^5(TT)_0$ sublevels. The ESR transitions of $^5(TT)_0 \rightarrow {}^5(TT)_{+1,-1}$ occur at the outer magnetic field strengths when the $B_0$ is perpendicular to the aromatic planes from the electron spin polarization imaging[41] in Supplementary Fig. 21B. On the other hand, the ESR intensity at the field strength at "X, Y" in Supplementary Fig. 22A is dominated by the $^5(TT)_0 \rightarrow {}^5(TT)_{+1,-1}$ resonances for $B_0$

directing to the pentacene backbone (Supplementary Fig. 21A). The isolated triplet is produced by the ISC of a single pentacene unit, which also agrees well with the MD simulation and the TAS results. On the other hand, even a few microseconds after the generation of $^5(TT)$, no $^3(TT)$-like ESR signal was observed, and the triplet retained its ISC-derived spectral shape. The ISC-derived triplet and the $^3(TT)$ signal are expected to be observed at similar magnetic field, but the single-exponential decay of the triplet signal suggests that no observable amount of $^3(TT)$ was generated (Supplementary Fig. 23). The absence of $^3(TT)$ was also supported by the calculated matrix elements of the absolute magnitudes of the spin Hamiltonian based upon the TT geometry (Supplementary Table 3). The ESR spectra in Fig. 4 did not show the characteristic A/E/A/E spin polarization pattern of the triplet dissociated from $^5(TT)_0$[42], suggesting that the contribution of the dissociated triplet is almost negligible. This is consistent with the DLS and MD simulation results that NaPDBA forms dimers in water-glycerol and does not form larger aggregates. The polarization lifetime of the quintet of NaPDBA aggregates and the NaPDBA-γCD complex were 5.5 and 1.8 µs, respectively (Supplementary Fig. 23D, F), and was determined by the deactivation of the multiexcitons as observed in the ns-TAS (Supplementary Fig. 17D). This time constant is sufficient to transfer the polarization to the nuclear spins by microwave irradiation. It is notable that the SF-derived polarization was long enough in the

water-glycerol glass, the most important matrix towards the biological applications of DNP. The ESR spectrum of the NaPDBA-βCD complex showed only the ISC-derived triplet signal and no quintet signal (Supplementary Fig. 23B).

## DNP using quintet electron spin polarization

DNP was then performed using the polarized electron spins produced in these supramolecular assemblies. The integrated solid effect (ISE) sequence was used for the DNP experiments (see Fig. 1B, C and Supplementary Fig. 24)[31,43]. The sequence starts from laser irradiation to produce the polarized electron spins. Then, microwave and magnetic field sweep are applied simultaneously. By matching the Rabi frequency of electron spins and the Larmor frequency of nuclear spins (proton in this experiment) under the microwave irradiation, the electron spin polarization can be transferred to nuclear spins. This is called as Hartmann–Hahn condition ($\gamma_e B_1 = \gamma_H B_0$), where $\gamma_e, \gamma_H$ is the gyromagnetic ratio of the electron and proton spin, respectively. $B_0$ is the external magnetic field, and $B_1$ is the oscillating magnetic field by microwave irradiation. Note that $B_1$ is applied perpendicular to $B_0$, and proportional to the square root of microwave power. Due to the broadening of ESR linewidth by hyperfine coupling and different molecular orientations, only a small fraction of spin packets can satisfy the Hartmann-Hahn condition at a time. One of the solutions of this problem is to use the ISE sequence that sweeps the external magnetic field during the microwave irradiation, so that more spin packets can be used for the polarization transfer. The transferred polarization is automatically diffused throughout the sample by dipolar interaction between proton spins. By repeating the above sequence, the proton polarization builds up while balancing with the spin-lattice relaxation.

The samples were prepared in a similar way as for the TAS and ESR measurements using a 5-mm ESR tube. To selectively polarize protons of water molecules and extend the spin-lattice relaxation time of $^1$H, we used a solvent mixture of deuterated glycerol, $D_2O$, and $H_2O$ in the volume ratio 5:4:1. The DNP sequence shown in Fig. 1C was performed with a central magnetic field of 629 mT, the field corresponding to the quintet ESR peak. The sweep range of the magnetic field was about 10 mT (Supplementary Fig. 25). We note that different magnetic fields were used for ESR and DNP. ESR was measured at a resonance frequency of 9.0 GHz to compare with common X-band data, while DNP was measured at 17.3 GHz, the frequency at which microwave amplifiers are available as high as possible to increase the $T_1$ of $^1$H. Remarkably, an increase in the $^1$H NMR signal intensity was clearly observed for the NaPDBA-only assemblies and the NaPDBA-γCD complex at a position corresponding to the $^5(TT)_0 \rightarrow {}^5(TT)_1$ ESR peak (Fig. 5A–D and Supplementary Fig. 26). However, this enhancement was not observed in the NaPDBA-βCD complex, which did not show the quintet-derived signal. Since ISE sequence was carried out for 10 μs, the ESR spectrum integrated for 10 μs after photoexcitation was compared to the DNP profile. With NaPDBA and NaPDBA-γCD, signal enhancement of 30% and 46% was observed in the magnetic field (629 mT) corresponding to the quintet ESR peak compared to the magnetic field of 641 mT for NaPDBA and 639 mT for NaPDBA-γCD corresponding to the triplet ESR peak, respectively (Fig. 5C, D). On the other hand, with NaPDBA-βCD, this ratio was remarkably low (7%), supporting the absence of quintet-induced DNP (Fig. 5E). The emissive enhanced NMR peaks were observed by using emissive quintet ESR peak for NaPDBA and NaPDBA-γCD. Meanwhile, no emissive NMR peak was observed for NaPDBA-βCD at the magnetic field corresponding to the emissive quintet ESR peak. There was an approximate correlation between ESR and DNP profiles, confirming that DNP was performed using triplets and quintets, respectively. Although triplet gave the larger DNP effect than quintet in the present system, it is possible to suppress the formation of triplet by selectively synthesizing dimers, and it would be possible to improve the quintet DNP performance more by selectively generating $^5(TT)_0$ among the quintet sublevels by

controlling the orientation of the dimers relative to the magnetic field[21]. These results indicate that nuclear hyperpolarization of water molecules was successfully achieved using SF-derived quintet electron polarization.

The difference in the obtained $^1$H polarization enhancement depends on several factors including the generation efficiency and polarization ratio, the polarization lifetime, and the spin-lattice relaxation of the nuclear spins around the polarizing agents. In the NaPDBA-only sample, there was a 20-fold enhancement at the quintet ESR peak (Fig. 5A). A smaller enhancement of 6.5-fold was observed for the NaPDBA-γCD complexes (Fig. 5B). The signal-to-noise ratio of the ESR spectra of NaPDBA and NaPDBA-γCD is very different, which suggests that the ESR intensity mainly affects the enhancement factor, but the difference in polarization lifetime may also have an effect. The lower enhancement factor of the NaPDBA-γCD complex was probably due to the triplet deactivation caused by triplet-triplet annihilation (TTA) between neighboring pentacenes, as observed in ns-TAS (Supplementary Fig. 17). The fact that the ESR could be fitted using similar rate constants also supports this inference. This indicates the importance of proper arrangement and interaction between pentacene units. Further support for this is that the build-up time was slowed down to the same level as the spin-lattice relaxation time only for the NaPDBA-γCD complex (Supplementary Figs. 27 and 28). This suggests that not enough electron spin polarization was produced for DNP, as shown by the lower ESR signal-to-noise ratio compared with the NaPDBA-only sample, showing that many triplets are deactivated in the strongly interacting pentacene dimer (Supplementary Fig. 23A, C).

We then checked the dependence of the microwave intensity irradiated during the DNP sequence using the quintet and triplet electron spin polarizations (Fig. 5F). The NMR signal was found to be maximized at a weaker microwave intensity with the quintet than with the triplet. This is because the Rabi frequency of electrons in the quintet state is $\sqrt{3}$ times higher than that in the triplet state[17], and the Hartmann–Hahn condition is accordingly shifted. This result further confirmed that the SF-derived polarized quintet state is successfully utilized for the DNP application.

In this study, we have demonstrated the potential of SF as a "polarized spin generator" in DNP of water molecules. We have created three supramolecular systems with different interactions between pentacene units using the identical complex of NaPDBA with β/γCD. SF was observed for the NaPDBA-only assemblies and the NaPDBA-γCD complex in water-glycerol glass, where the pentacene skeletons are close to each other. The promising potential of SF as a polarization source was indicated by the preferential population of the $|^5(TT)_0\rangle$ sublevel at a particular resonance magnetic field strength. The magnetic field dependence and microwave intensity dependence of DNP ensure that nuclear polarization was enhanced using quintet-derived polarization. Our findings indicate that the enhancement factor of DNP varies with assembly structure, indicating the direction of further optimizations. The remaining challenge for selective formation of the $^5(TT)_0$ state is to orient the chromophores with respect to the magnetic field, which would be achieved by using an orienting agent or by orienting dispersed nanocrystals[44,45]. Until now, the application of SF has been mainly limited to the field of energy. This study opens a new potential application of SF to quantum biotechnology, and will accelerate the development of quantum sensing and quantum information science based on the unique multiexcitons exhibited by organic chromophores.

## Methods
### Materials
All reagents and solvents for measurements were used as received without further purification. Glycerol was purchased from Kishida chemical. Sodium 3-(trimethylsilyl)−1-propanesulfonate was purchased

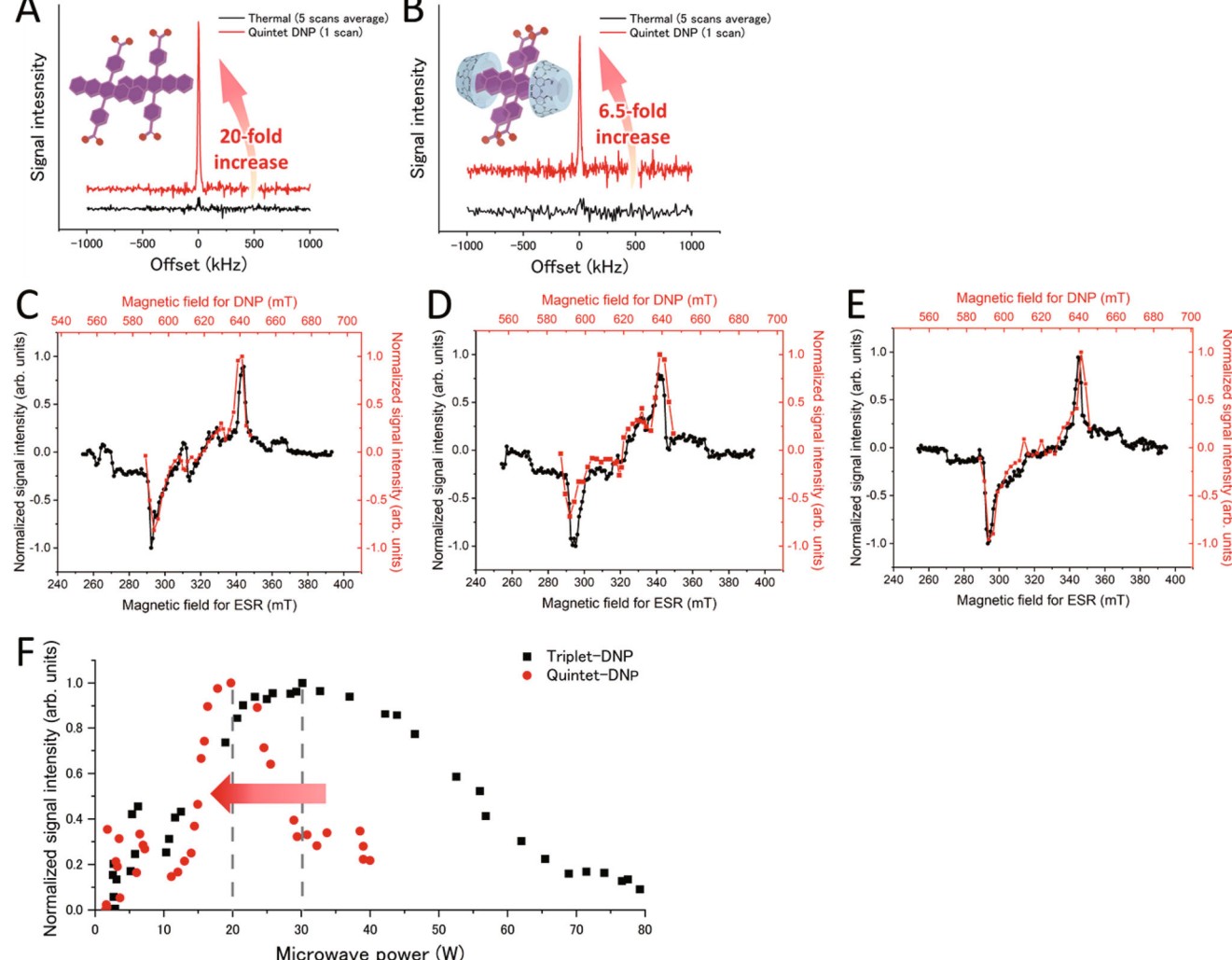

**Fig. 5 | DNP using SF-born quintet electron spin polarization. A, B** $^1$H-NMR signals under thermal conditions (black lines, 5 scans every 10 min) and after quintet-DNP (red lines, ISE sequence for 5 min, 1 scan) of water-glycerol (glycerol-$d_8$:D$_2$O:H$_2$O = 5:4:1) containing **A** NaPDBA and **B** NaPDBA-γCD at 100 K ([NaPDBA] = 1 mM, [γCD] = 5 mM). The photo-excitation wavelength and frequency were 527 nm and 500 Hz, respectively. DNP was performed by matching the magnetic field to the quintet peaks. The microwave power and frequency were 20 W and 17.30 GHz, respectively, the laser powers were 2.7 W for **A** and 1.5 W for **B**. The magnetic field sweep width was 10 μs. Magnetic field dependence of the signal intensity of the $^1$H NMR by DNP and time-resolved ESR spectra in water-glycerol containing **C** NaPDBA, **D** NaPDBA-γCD, and **E** NaPDBA-βCD ([NaPDBA] = 1 mM, [βCD] = [γCD] = 5 mM). Water-glycerol glass (glycerol-$d_8$:D$_2$O:H$_2$O = 5:4:1) was used for the DNP measurement at 100 K. ISE sequence for 20 s (**C**, **E**) and 30 s (**D**) and 1 scan;

microwave power and frequency were 20 W and 17.25 GHz, respectively; laser power: 1.5 W; magnetic field sweep width: 10 μs. Water-glycerol glass (glycerol:H$_2$O = 5:5) was used for the time-resolved ESR measurements at 143 K. ESR spectra were integrated for 10 μs after photoexcitation in order to compare the DNP profile with the ISE sequence for 10 μs. **F** Microwave power dependence of DNP enhancement. The gray dashed line indicates the microwave power when the $^1$H-NMR signal is at its maximum. The red arrow indicates the peak top shift from triplet-DNP to quintet-DNP. Triplet-DNP was performed at 27.4 MHz (ISE sequence for 10 s and 4 scans with a laser power of 2.7 W, microwave frequency of 17.30 GHz and sweep width of 25 μs). Quintet-DNP was performed at 26.9 MHz (ISE sequence for 10 s and 10 scans with a laser power of 2.7 W, microwave frequency of 17.30 GHz and sweep width of 10 μs).

from TCI. βCD and γCD were purchased from Wako Pure Chemical. Deuterium oxide and glycerol - $d_8$ were purchased from CIL. Deionized water was generated by Direct-Q UV (Merck Millipore). The synthesis and characterization of H$_2$PDBA have been reported in our previous work[46]. Analytical grade methanol and sodium hydroxide purchased from Wako Pure Chemicals were used to form the sodium salt NaPDBA in methanol, and the solvent was removed to form NaPDBA.

### General characterizations
$^1$H NMR (400 MHz) spectra were measured on a JEOL JNM-ECZ 400 and Bruker Ascend NMR 400 MHz using Sodium 3-(Trimethylsilyl)−1-propanesulfonate as the internal standard. UV-vis absorption spectra were measured by JASCO V-670 and V-770 spectrophotometers. Dynamic

light scattering (DLS) was carried out using a DLS-8000DL (Otsuka Electronics).

### Pump-probe transient absorption
The fs-TAS and ns-TAS were conducted using home-built pump-probe setups. The light source was a Ti:sapphire regenerative amplifier (Spectra-Physics, Spitfire Ace, pulse duration: 120 fs, repetition rate: 1 kHz, pulse energy: 4 mJ/pulse, central wavelength: 800 nm) seeded by a Ti:sapphire femtosecond mode-locked oscillator (Spectra-Physics, Tsunami, pulse duration: 120 fs, repetition rate: 80 MHz, pulse energy: 10 nJ/pulse). The output of the amplifier was divided into two pulses. One of the outputs was led to a phase-matched BBO crystal for second harmonic generation and was used for the pump pulse

(400 nm, 120 fs). The pump pulse was focused on the sample solution in a 1-mm path length quartz cell. The other output was focused on a sapphire crystal (3 mm thickness), and generated white light (450–750 nm) for the probe pulse. The angle between the pump and probe polarizations was set to the magic angle (~54.7 deg.). The probe pulse that passed through the sample solution was dispersed by a polychromator (JASCO, CT-10, 300 grooves/500 nm), and the spectra were recorded by a multichannel detection system with a CMOS sensor (UNISOKU, USP-PSMM-NP).

The TAS measurements on the water-glycerol system were conducted under cryogenic temperature conditions. The glassy sample was prepared by the rapid cooling procedure and was held in a cryostat for spectroscopy (UNISOKU, USP-203 Series) during the measurements. The output from the regenerative amplifier was led to an optical parametric amplifier system (Light Conversion, TOPAS-prime) to generate 500 or 600 nm pulses for the pump pulse of the fs-TAS. The output pulse from a nanosecond optical parametric oscillator system (EKSPLA, NT220) was employed for the pump pulse with a wavelength of 500 or 600 nm for the ns-TAS. We chose the lower excitation photon energy to suppress excess energy and transient heating in the excitation process because the glassy state was crystalized when too much heat was caused by photoexcitation. The probe white light was generated from the output of the Ti:sapphire amplifier focused on a sapphire crystal. The probe pulse passed through the sample solution was dispersed by a polychromator (JASCO, CT-10, 300 grooves/500 nm), and the spectra were recorded by a multichannel detection system with a CMOS sensor (UNISOKU, USP-PSMM-NP). The recorded data were analyzed using a home-build program based on Python.

### Time-resolved ESR

The time-resolved ESR measurement was performed on a home-built spectrometer (Supplementary Fig. 29), which has been described in our previous report[47].

The samples were inserted into the dielectric resonator inside of an electromagnet (MC160-60G-0.8 T, Takano Original Magnet) which is controlled by a function generator (33500B, Keysight). The sample was photo-excited by using a pulsed laser (Tolar-527, Beamtech Optronics). The pulse width, maximum repetition rate and maximum power of this laser are 200 ns, 5 kHz and 400 W, respectively. For samples, the repetition rate and the power of the laser were set to 100 Hz and 0.3 W.

A microwave was generated with the power of ~10 µW (SG24000H, DS Instruments) and amplified by using a power amplifier (ALN0905-12-3010, WENTEQ Microwave Corp), then converted to DC with a diode detector (DHM185AB, Herotek). ESR signal was also amplified and the noise was cut off by using two amplifiers (SA-230F5, NF ELECTRONIC INSTRUMENTS and 5305 differential amplifiers, NF ELECTRONIC INSTRUMENTS). The ESR signal was detected by an oscilloscope (DSOX3024T, Keysight). The temperature was controlled by flowing cold nitrogen gas into the microwave cavity.

The dielectric resonator was fabricated as shown in Supplementary Fig. 30. A ring-shaped dielectric ceramic (M29, MARUWA) with an outer diameter of 6.8 mm, an inner diameter of 2.1 mm, and a thickness of 2.5 mm was used. The hole with diameter of 14 mm was made on the copper blocks and closed with two copper clad laminate (CCL) boards to insert the dielectric ceramic. There is a hole with diameter of 2.1 mm on the top side CCL boards to insert the sample, and there are three holes on the center of front, back and side of copper blocks for microwave irradiation, laser irradiation and flowing the gas for temperature control with diameter of 8, 6, and 5 mm, respectively. Two dielectric ceramics were hold by PTFE in the center of resonator. The waveguide was attached to the front of the resonator and a small copper plate, placed between the waveguide and the resonator, was controlled by a PTFE screw to adjust the

microwave reflectivity. The resonance frequency of the fabricated cavity resonator was 9 GHz.

ESR spectra were analyzed in Matlab version R2019b Update 8 (The Mathworks, Inc.) using the computational model shown in the previous report[16]. The polarization of the triplet state was assumed to be generated by only spin–orbit ISC, suggesting that no triplet dissociations occurred from the dimer. The spectra derived from quintet state were simulated by two strongly coupled TT conformations ($TT_A$ and $TT_B$) which has different orientations and $J$-coupling. The fitting parameters are summarized in Supplementary Fig. 31 and Supplementary Tables 1 and 2.

### DNP experiment

DNP experiment was carried out on a home-built spectrometer (Supplementary Fig. 32), which has been described in our previous report[47]. It consists of electromagnet (MC160-60G-0.8 T, Takano Original Magnet), microwave resonator, coil for magnetic field sweep and pulsed laser. Sequence control and NMR signal detection were performed using the OPENCORE NMR spectrometer[48].

The cavity was fabricated as shown in Supplementary Fig. 33. The hole with diameter of 21.5 mm was made on the copper blocks and closed by two CCL boards to adjust the resonant frequency of Ku-band. There is a hole with diameter of 5 mm on the top side CCL boards to insert the samples, and there are three holes on the center of front, back and side of copper blocks for microwave irradiation, laser irradiation and flow of cold nitrogen gas to keep the sample temperature, with diameter of 7 mm, 6 mm and 5 mm, respectively. The microwave reflectivity was adjusted by a Teflon screw as with time-resolved ESR setup. The resonance frequency of the fabricated cavity resonator was about 17.3 GHz. One-turn saddle coil, four enamel wires, which were connected to each other, were inserted into the resonator to surround the sample. An induced magnetic field was applied to the sample by a current flowing in the copper wire, and the induced field direction is parallel to the static magnetic field. The magnetic field was swept from a high field to a low field over a few tens of microseconds. The coils for the NMR detection were installed on the top of the resonator. The coil was made by winding enameled wire. The oscillating magnetic field from the coil and the static magnetic field from the electromagnet were arranged perpendicular to each other.

The sample was photo-excited by using pulsed laser (Tolar-527, Beamtech Optronics) and kept the temperature by flowing cold nitrogen gas, same as time-resolved ESR setup. For triplet-DNP experiments, the repetition rate and the power of the laser were set to 500 Hz and 1.5–2.7 W. The continuous microwave was generated (SG24000H, DS Instruments) and converted to a pulsed wave using a pin diode (S1517D, L3HARRIS). The pulsed wave was amplified by using a power amplifier (AMP4081P-CTL, EXODUS ADVANCED COMMUNICATIONS) and sent into the resonator by a coaxial cable with the transmission loss of ~1 dB. The magnetic field sweep was performed by applying an amplified triangular wave to a copper wire built in the resonator (Supplementary Fig. 33). The source triangular wave was generated from the function generator (WF1974, NF ELECTRONIC INSTRUMENTS). This triangular wave was amplified tenfold using an operational amplifier (137-PA05, Apex Microtechnology) and applied to the copper wire to reach a maximum of ±50 V. NMR signals were obtained by OPENCORE NMR spectrometer. The solenoid coil was used as NMR probe coil and mounted on the top of resonator (Supplementary Fig. 33). The sample was rifted up to the probe coil by using stepping motor within 1 s before NMR detection. In the case of protons in solid samples, a magic echo sequence was used because the short $T_2$ relaxation time and strong dipole interaction make it difficult to detect them with ordinary single pulses or spin echoes. The sweep width of the magnetic field when performing the ISE sequence is shown in SI. According to Supplementary Fig. 25, a voltage application of 1 V generates a magnetic field of 0.2 mT from the field sweep circuit. Thus, a 50 V sweep in the ISE

sequence in this study corresponds to a 10 mT sweep. This value is narrow enough to selectively use only one ESR peak for DNP.

## Molecular dynamics simulations

All-atom MD simulations in this study were performed by using the MD program GROMACS 2016.3. In the initial structure of the systems of NaPDBA in water:glycerol = 1:1 ([NaPDBA] = 1 mM, [βCD] = [γCD] = 5 mM), the complexes of NaPDBA with β/γCD or NaPDBA molecules were assembled close to each other and solvents and isolated β/γCD were placed in the surrounding space to fill the cubic MD cell. For the system of NaPDBA in water ([NaPDBA] = 1 mM, [βCD] = [γCD] = 5 mM), the complexes and isolated β/γCD molecules were randomly inserted in the MD cell filled with water molecules. The number of molecules in each system is listed in Supplementary Table 4. The generalized Amber force field[49] parameters were used for the force field parameters of NaPDBA, βCD, glycerol, and γCD and the TIP4P-Ew[50] model was used for the water molecules. As NaPDBA is composed of sodium ion and PDBA⁻, their partial atomic charges were separately assigned. The atomic charges of PDBA⁻, βCD, and γCD were calculated using the restrained electrostatic potential (RESP)[51] methodology, based on DFT calculations (B3LYP/6-31G(d, p)) using the GAUSSIAN 16 revision C01 program (Gaussian, Inc., Wallingford CT, 2016).

In the present MD simulations, pre-equilibration and equilibration runs at room temperature (300 K) were sequentially carried out after the steepest energy minimization. The pre-equilibration and equilibration for the 2:2 inclusion complex of NaPDBA-γCD in water-glycerol at a lower temperature (243 K) were performed after the equilibration at room temperature. During the 5 ns preequilibration, the temperature and pressure of the system were kept constant using Berendsen thermostat and barostat[52] with the relaxation times of 0.2 and 2.0 ps, respectively. The equilibration was run for 20 ns using the Nosé–Hoover thermostat[53] and Parrinello–Rahman barostat[54] with the relaxation times of 1.0 and 5.0 ps, respectively. The pressure of the system for all MD simulations was kept at 1 bar. All bonds connected to hydrogen atoms were constrained with LINCS[55] algorithm. The time step of preequilibration and equilibration was set to 2 fs. The long-range Coulomb interactions were calculated with the smooth particle-mesh Ewald method[56] with a grid spacing of 0.30 nm. The real space cutoff for both Coulomb and van der Waals interactions was 1.2 nm.

The PMFs for pulling away one of γCD forming the complex with NaPDBA were calculated for the quantitative comparison of the stability of the complex in water-glycerol between NaPDBA-γCD at 300 K and 243 K. According to the previous study[57], the PMF for each system was obtained from a series of umbrella sampling (US) simulations, where the energy minima of their umbrella potentials were located at equal intervals along the direction of pulling γCD and the calculated probability density distributions were overlapped. For preparing the initial positions of two series of the US simulations, nonequilibrium steered MD simulations were performed using the structure after the 20 ns equilibration runs. The center of the mass of one molecule of γCD molecules forming the complex was pulled away in one direction using an umbrella potential with a force constant of 1000 kJ/mol/nm². The pulling rate was 0.35 and 0.80 nm/ns for NaPDBA-γCD in water-glycerol at 300 and 243 K, respectively. During the 3 ns steered MD simulations using the Nosé–Hoover thermostat and Parrinello-Rahman barostat, the atoms of NaPDBA except sodium ions and hydrogen atoms were constrained to the initial positions by a harmonic potential with a force constant of 1000 kJ/mol/nm². In the US simulation with the structures selected from the trajectory of the steered MD run, the 1 ns preequilibration run using the Berendsen thermostat and barostat and the 2 ns equilibration run using the Nosé–Hoover thermostat and Parrinello–Rahman barostat were performed. The number of the US simulations was 6 both for NaPDBA-γCD at 300 K and at 243 K. The force constant of the umbrella potential used in the US simulations was also 1000 kJ/mol/nm². The values of the PMF were calculated from

each set of the US simulations by the weighted histogram method (WHAM)[58,59].

## Data availability

The raw data generated in this study have been deposited in the Zenodo https://doi.org/10.5281/zenodo.7554559 and available from the corresponding authors upon request.

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

## Acknowledgements

The computations were partially performed at the Research Center for Computational Science, Okazaki, Japan (Project: 21-IMS-C043, 22-IMS-C043). This work was partly supported by the JST-PRESTO program on "Creation of Life Science Basis by Using Quantum Technology" (JPMJPR18GB); JST-FOREST Program (JPMJFR201Y); JSPS KAKENHI (JP17H06375, JP19H02537, JP19H05718, JP19K15508, JP20H05106, JP20H02713, JP20K21211, JP22K19051, JP20H05676, JP19H00888, JP20H05831, JP22J21293, and JP21J13049); JST SPRING (JPMJSP2136); JST, the establishment of university fellowships towards the creation of science technology innovation (JPMJFS2132); The Shinnihon Foundation of Advanced Medical Treatment Research; the Innovation Inspired by Nature Program of Sekisui Chemical Co. Ltd.; the RIKEN-Kyushu

University of Science and Technology Hub Collaborative Research Program; the RIKEN Cluster for Science, Technology and Innovation Hub (RCSTI); and the RIKEN Pioneering Project "Dynamic Structural Biology"; Kyushu University Platform of Inter-/Transdisciplinary Energy Research (Q-PIT) through its "Module-Resarch Program."

## Author contributions

Y. Kawashima and N.Y. conceived and designed the project. Y. Kawashima, Y.N., and S.F. prepared and characterized the samples. S.S and G.W. performed the MD simulations. T.R., T.T., M.S., K.O., and K.M. carried out the TAS measurements. Y. Kawashima and A.Y. measured the ESR spectra. A.Y. and Y. Kobori simulated the ESR spectra. Y. Kawashima, T.H., and K.N. carried out the DNP measurements. K.T. and T.U. contributed to building the ESR and DNP setup. Y. Kawashima, A.Y., K.N. N.K., K.M., and N.Y. wrote the manuscript with contributions from all authors.

## Competing interests

The authors declare no competing interests.
