## [Peer Review File · Nature Communications]

Singlet fission as a polarized spin generator for dynamic nuclear polarizationREVIEWER COMMENTS

Reviewer #1 (Remarks to the Author):

Overall, the DNP experiment involving singlet fission chromophores (both aggregated and bound by cyclodextrin) appears interesting and novel. The authors performed appropriate control experiments and analyzed their results to a sufficient depth to support their observation of successful DNP of the protons of water molecules using excitons generated from SF.

The abstract and introduction highlight the quintet, and particularly the 5TT0 state as most relevant for DNP via singlet fission. However, the results suggest that the quintet is far from the most dominant state produced, and 5TT0 is not the sublevel with the most population. I understand that highlighting the quintet improves the novelty aspect, but it is somewhat disingenuous to suggest it is the most efficacious. In fact, it is never mentioned in the main text (only deep within the SI) that the triplet shows much larger DNP enhancement than the quintet (70-fold to 20-fold). I would prefer the authors re-write the introduction with a more balanced tone. Indeed, singlet fission plays a role in producing DNP, but a dominant role for 5(TT)0 is not demonstrated.

This fact undercuts the claims of applicability and potentially favorable comparisons to other methods of DNP. For example, I'd be interested in seeing some investigation of the theoretical limits of using quintets for DNP. Is the maximum efficiency under ideal conditions better than currently existing agents? Could these ideal conditions be realistically achieved? Without such an analysis, the field will not likely be inspired by the results. The work could be published after some modifications, but without stronger support for broader impact, it may be more well-suited to a specialized journal.

I have a variety of comments and suggestions that should be addressed:

(1) There are frequent references to "selective" generation of 5TT0. This requires both intermolecular alignment, which is true for some of the dimers studied, and alignment with an applied magnetic field, which is not true for unoriented powder samples described in the paper and likely most relevant for DNP applications. The authors are likely aware of this, as the fitting of their spectra already involves population of $5TT\pm 2$, and it is just an issue of word choice. To me, "selective" suggests pure population of 5TT0, and "preferential" would be more accurate. However, the difference between 100% population of a desired state and a mixture of states, bound by physically relevant limitations, does undercut the argument that SF quintets are promising DNP agents.

(2) The yield of quintet in the supplied EPR spectra is low relative to that of ISC triplet in both the aggregated and CD-bound samples. Could this limitation be overcome?

(3) It doesn't seem obvious to me that lower MW power (approx. 2/3) required to maximize the DNP effect for quintets vs. triplets is inherently valuable. This claim is also undercut by the fact that the peak for DNP efficiency of the triplet is broad and appears to already be at 90% of its maximum at the peak power of the quintet.

(4) The section about aggregation and the associated absorption spectra seems inconsistent. The absorption shift observed for the NaPDBA is assigned to aggregation but not accompanied by peak broadening or splitting. By the dimer structures given, H aggregates should be formed, resulting in a blue shift. This is not observed. Why? The CD complex shows considerable broadening, suggesting multiple intermolecular species. Much of the interpretation of aggregates is based on the MD simulations, which may not capture the true picture. NOESY is performed, but exactly how those data are used to verify the aggregate models is not clear. It seems that DOSY would also be a useful method. The authors should do a better job of describing the connection between analytical methods and the aggregate structures shown.

(5) The pump-probe TAS section shows a straightforward production of TT, but the interpretation of the CD complex is overly exotic. It invokes three different sequentially population versions of TT. I don't understand how these are distinct electronically, especially considering that S1 and TT should not be close to isoenergetic. I think it is more likely that inhomogeneity is playing a role, leading to different timescales and spectra for TT formation. Do the conformers suggest for the EPR section play a role in the TA dynamics also?

(6) There is never discussion of the 3(TT) population, despite the possibility that it could form from 5(TT).

(7) This is probably understood by those working on DNP, but why is the magnetic field in Fig 1C modulated?

(8) What is the difference between the Euler angles and dipolar angles in Table S2? The caption refers to reference 57 in the main text, but there are only 42 references.

(9) Pg 9, lines 7-8—it should be said explicitly that the reason the orientation for which populations were calculated (field aligned with the molecular plane) would dominate because of the statistics of powder samples.

(10) Labeling of the two different dimers as TT1 and TT2 is confusing, as numerical subscripts and superscripts are used to denote spin multiplicity elsewhere. I would suggest TTA and TTB instead. I think there may also be a related mistake in labeling on pg. 10, line 13

(11) Abstract says “sole way” to generate quintet, text says “sole way without heavy metals”

(12) Pg 2, line 27 – DNP for medical applications isn’t typically grouped with the “second quantum revolution”

(13) Pg 2, line 28 – More detailed theoretical models (e.g. JDE model) indicate that true selective population of 5TT0 requires shared molecular axes and their alignment with B0

(14) Pg 3, line 5— it is stated that up to 200% quintet yield (where triplet is probably intended) can be achieved, but the reference cited does not even mention quintets

(15) Pg 3, line 19—Again using language that suggests selective population of 5TT0, when models and TR-EPR spectra in the paper suggest that’s not the case for an unoriented sample

(16) “Because the ESR spectral analysis showed a similar singlet recombination rate constant from the 1(TT) dimer, the lower enhancement factor of the NaPDBA- γ CD complex was probably due to the triplet deactivation¹⁶ caused by triplet-triplet annihilation (TTA) between neighboring pentacenes, as observed in ns-17 TAS.”

How was singlet recombination rate obtained from EPR? Is 1(TT) supposed to read TT1?

Reviewer #2 (Remarks to the Author):

This is a very interesting paper showing that singlet fission processes, which generate a non-equilibrium spin-2 quintet state, can be used to enhance nuclear polarization by DNP procedures. The experimental results are modest, showing NMR signal enhancements of around 20, but do provide intriguing evidence of a new family of DNP agents and procedures that could well be very useful in certain circumstances. Certainly the concept is novel and constitutes an impressive piece of science crossing over from photovoltaics to NMR, MRI, and eventual possible clinical applications.

The paper is, for the most part, very well-written. However there are few points of concern which, in my view, should be addressed.

* In the title, the abstract, and in many places in the text, the experiment is called “biological”. It isn’t. The method may hold some promise for eventual biological applications, but this is most definitely not “biological DNP”. I have to call this what it is: the gratuitous, and ill-advised, use of buzzwords, to enhance the chance of publication. There is nothing biological about the described experiments.

* on page 2, line 31, it is stated that the singlet fission “preserves its total spin”. This is manifest nonsense, since the singlet fission process goes from $S=0$ to $S=2$. The authors appear to mean that the magnetic quantum number m_S is preserved, at least approximately. That may well be true under certain circumstances, but the phrase should be reworded so that the science is accurate, and it should be indicated within which regime or set of approximations the m -value is preserved. Certainly this is not an absolute selection rule, which the authors appear to imply.

* on page 3, line 21, the “Hartmann-Hahn condition” is indicated. Whatever is going on in figure 1B and 1C, it is certainly not Hartmann-Hahn matching. In general, the description of the procedure for the transfer of electron spin-polarization to nuclear polarization is very poor and poorly referenced. This aspect of the paper needs to be greatly improved, in my opinion.

* on page 10, line 2, we read that “the magnetic field is aligned with the ESR peaks”. What is this supposed to mean? Needs rewording to be made scientific, clear, and rigorous.

* in the caption to Fig.5, the “ISE sequence” is mentioned. As far as I can tell this is unexplained.

In general, the description of the electron-to-nuclear polarization transfer procedure, and its operation, are very poor and need considerable improvement.

Reviewer #3 (Remarks to the Author):

The authors prepare pentacene derivative (NaPDBA) supramolecular assemblies using beta or gamma cyclodextrin in water or water-glycerol matrices and report the transient EPR and transient absorption spectra of these materials. The prepared NaPDBA complexes are found to favor NaPDBA monomers or multimers (including dimers) as is evidenced by presence of singlet fission (SF) in some of these materials which would require a close spatial proximity between two of the NaPDBA molecules and the absence of SF in the monomer cases. The optical excitation of these materials produce both a polarized

triplet (via ISC) and polarized quintet state (via ISC/SF) and the authors demonstrate that this optically generated spin polarization from both the triplet and quintet states may be used to enhance ^1H NMR spectra via ISE DNP. To my knowledge, this work would be the first demonstration of polarization transfer between a polarized quintet electron ensemble and ^1H via ISE DNP. The original study is well done and thorough, with experimental results supported by molecular dynamics simulations. This study would be of interest to a broad audience interested in material science, magnetic resonance and fundamental spin physics. The work is also of significance to the fields of hyper polarized NMR and quantum information science. I would recommend that this paper be published after revisions and following the authors' answers to the following comments and questions:

General Comments and Questions:

1. The statement that singlet fission is the sole way to generate a spin quintet state in organic materials is not true. Please see the following publications in which quintet ground states (Q0) are formed through photolysis in ketocarbenes <https://pubs.acs.org/doi/pdf/10.1021/ja00353a055> and non-Kekulé structures <https://pubs.acs.org/doi/pdf/10.1021/ja00212a006>. Please address this by revising your statements throughout the text. Quintets (Q1) in organic materials are also formed in triplet-triplet annihilation upconversion, albeit they are less stable. Perhaps the authors meant to say that long lived Q1 only accessible via SF? Or polarized Q1? Please clarify this point.
2. Not all experimental details were included. What was the sweep range (in mT) used for the magnetic field for polarization transfer for the triplet DNP? What about for the quintet DNP? The authors only report on the sweep rate, not the range. The range should be reported for all of the DNP experiments.
3. The authors report that the lower enhancement factor of the NaPDBA gamma CD complex is likely due to triplet deactivation, however, a simpler explanation would be that the polarization lifetime of the NaPDBA-gammaCD complex is nearly 3 times shorter than that of the NaPDBA aggregates, wouldn't this be the main contribution to the decreased enhancement? Especially given that the length of each DNP cycle is longer than 10 microseconds?
4. The authors mention that an enhancement at the quintet position was not observed for the NaPDBA-betaCD complex. Could the authors include this data in figure 5? As this would be the most convincing evidence that the polarization is derived from the quintet.
5. Unless the sweep range for the ISE (mT) is very narrow about the quintet absorption peak < 10 mT, wouldn't the authors expect some polarization contribution from the triplet? Please clarify this point. Did the authors also measure the DNP experiment using the emissive quintet peak to observe an

inverted ¹H signal? It would be interesting to see the full DNP profile (if the authors have the data) and see how well it matches with the trEPR spectrum.

6. Figure S11. The authors should discuss the complexation energies obtained through MD rather than just showing a region on a snapshot of the MD, as these calculated values would be more informative to support the observed trEPR and TA data.

7. Figure S13. What was the excitation energy and why was a different excitation energy (rather than 600 nm) used for this dataset?

8. Figure S15. Why is no triplet observed in this figure for NaPDBA-betaCD? This does not match S14.

Formatting and grammar corrections:

1. The order in which the supplementary figures are introduced in the main text is not in order, this makes it time consuming to navigate between the main text and supplementary. Please place figure labels in order as they are introduced.

2. Figures 3A and 3F would be more informative if the figure is shown from $x = 450$ to 550 nm as the features discussed in the caption cannot be seen clearly as it is currently shown.

3. Figure 4: Please show the quintet manifold total fit (as you have for the triplet - dotted line) as it is difficult to see the quality of the fit as it currently is.

4. Typo page 10 line 8, chemical

5. Typo page 10 line 36, glassy

6. Typo page 14 line 12, were

Replies to the comments of Reviewer 1:

Overall, the DNP experiment involving singlet fission chromophores (both aggregated and bound by cyclodextrin) appears interesting and novel. The authors performed appropriate control experiments and analyzed their results to a sufficient depth to support their observation of successful DNP of the protons of water molecules using excitons generated from SF.

The abstract and introduction highlight the quintet, and particularly the $^5\text{TT}_0$ state as most relevant for DNP via singlet fission. However, the results suggest that the quintet is far from the most dominant state produced, and $^5\text{TT}_0$ is not the sublevel with the most population. I understand that highlighting the quintet improves the novelty aspect, but it is somewhat disingenuous to suggest it is the most efficacious. In fact, it is never mentioned in the main text (only deep within the SI) that the triplet shows much larger DNP enhancement than the quintet (70-fold to 20-fold). I would prefer the authors re-write the introduction with a more balanced tone. Indeed, singlet fission plays a role in producing DNP, but a dominant role for $^5(\text{TT})_0$ is not demonstrated.

This fact undercuts the claims of applicability and potentially favorable comparisons to other methods of DNP. For example, I'd be interested in seeing some investigation of the theoretical limits of using quintets for DNP. Is the maximum efficiency under ideal conditions better than currently existing agents? Could these ideal conditions be realistically achieved? Without such an analysis, the field will not likely be inspired by the results. The work could be published after some modifications, but without stronger support for broader impact, it may be more well-suited to a specialized journal.

I have a variety of comments and suggestions that should be addressed:

We are very pleased that the reviewer found our DNP using SF to be interesting and novel. We believe that the significance of our work lies in the novelty of the first application of DNP in water, focusing on the spin degree of freedom of SF.

On the other hand, as pointed out by the Reviewer, there is room for improvement in the selectivity of $^5(\text{TT})_0$ production, and we have changed the description to a more balanced tone, mainly in the introduction.

We also agree that the theoretical limit of quintet for DNP pointed out by the Reviewer is an important point. The triplet state of pentacene, which has been used as the best polarizing agent, has a polarization of about 70%. On the other hand, quintet produced by SF has the potential to theoretically exceed the performance of triplet, since it has

been reported that $^5(\text{TT})_0$ is selectively produced with 100% polarization in TIPS-tetracene thin film (Weiss, L. R. et al, *Nat. Phys.* **13**, 176-181, doi:10.1038/nphys3908 (2016)). Moreover, it is notable that this selective $^5(\text{TT})_0$ formation was achieved in randomly-oriented thin film. This means that $^5(\text{TT})_0$ can be selectively formed independent of molecular orientation relative to the magnetic field, which is very useful for DNP of biology-relevant molecules in water-based glass matrices.

We have included this discussion in the revised manuscript (page 13, line 26-33).

(1) There are frequent references to “selective” generation of $^5\text{TT}_0$. This requires both intermolecular alignment, which is true for some of the dimers studied, and alignment with an applied magnetic field, which is not true for unoriented powder samples described in the paper and likely most relevant for DNP applications. The authors are likely aware of this, as the fitting of their spectra already involves population of $^5\text{TT}_{\pm 2}$, and it is just an issue of word choice. To me, “selective” suggests pure population of $^5\text{TT}_0$, and “preferential” would be more accurate. However, the difference between 100% population of a desired state and a mixture of states, bound by physically relevant limitations, does undercut the argument that SF quintets are promising DNP agents.

We agree with the Reviewer that “preferential” is a more appropriate adjective for our system, since levels other than $^5(\text{TT})_0$ are also generated. We have revised the manuscript accordingly.

In our current system, the $^5(\text{TT})_0 \rightarrow ^5(\text{TT})_1$ transition peak of the pentacene dimer in a specific orientation was used for DNP at the magnetic field of 629 mT. In this orientation, other states such as $^5(\text{TT})_2$ and $^5(\text{TT})_{-2}$ are also populated, however, ESR transitions involving $^5(\text{TT})_2$ and $^5(\text{TT})_{-2}$ have almost no contribution at around 629 mT. To demonstrate this point, we computed EPR spectrum of ^5TT state obtained by the powder pattern calculation with considering the spin populations in all the sublevels $^5(\text{TT})_{+2}$, $^5(\text{TT})_{+1}$, $^5(\text{TT})_0$, $^5(\text{TT})_{-1}$ and $^5(\text{TT})_{-2}$ (Fig. S22A) and the powder pattern with only by $^5(\text{TT})_0 \rightarrow ^5(\text{TT})_{+1}$ and by $^5(\text{TT})_0 \rightarrow ^5(\text{TT})_{-1}$ contributions (Fig. S22B). At the field strengths represented by “Z” and “X, Y” in Fig. S21A, the EPR transition intensities (transverse magnetizations) by the quintet states are dominated by the resonances from the $^5(\text{TT})_0$ sublevels. This also means that the transitions by $^5(\text{TT})_{+2} \rightarrow ^5(\text{TT})_{+1}$ and by $^5(\text{TT})_{-2} \rightarrow ^5(\text{TT})_{-1}$ almost do not contribute to these transitions. Thus, the EPR intensity at the field strength at “X, Y” in Fig. S21A is dominated by the $^5(\text{TT})_0 \rightarrow ^5(\text{TT})_{+1}$ and by $^5(\text{TT})_0 \rightarrow ^5(\text{TT})_{-1}$ resonances.

We have included this discussion in the revised manuscript (page 9, line 7-19).

Fig. S22. (A) Computed EPR spectrum of ${}^5\text{TT}$ state obtained by the powder pattern calculation with considering the computed spin sublevel populations in ${}^5(\text{TT})_{+2}$, ${}^5(\text{TT})_{+1}$, ${}^5(\text{TT})_0$, ${}^5(\text{TT})_{-1}$ and ${}^5(\text{TT})_{-2}$ in (D) both of TT_A and TT_B states, by using the reported method¹. (B) Computed ESR spectrum of ${}^5\text{TT}$ state obtained by the powder pattern calculation only by ${}^5(\text{TT})_0 \rightarrow {}^5(\text{TT})_{+1}$ and by ${}^5(\text{TT})_0 \rightarrow {}^5(\text{TT})_{-1}$ contributions in (D). (C) Experimental ESR spectrum of NaPDBA (black line) and simulated ESR spectrum (green line) composed of the blue line in (A) and of the isolated triplet state generated by the ISC. (E) Conformations of the TT_A and TT_B states undergoing the mutual J -modulation between $J_\text{A} = -5.0$ T and $J_\text{B} = -0.8$ T in the TT_A and TT_B , respectively in the exchange-coupling (J). This model is used for the present calculations in the spin sublevel populations and the magnetizations.

Furthermore, it has been reported that ${}^5(\text{TT})_0$ is generated 100% selectively in randomly oriented films as described above (Weiss, L. R. et al, *Nat. Phys.* **13**, 176-181, doi:10.1038/nphys3908 (2016)). In other words, by forming a multimer larger than dimer, it is possible to form ${}^5(\text{TT})_0$ with 100% polarization independent of orientation, indicating that the concept of applying SF to DNPs has extremely high potential.

This discussion was added in the revised manuscript (page 13, line 26-33).

(2) The yield of quintet in the supplied EPR spectra is low relative to that of ISC triplet in both the aggregated and CD-bound samples. Could this limitation be overcome?

We appreciate this comment. As mentioned in the original manuscript, the system was a mixture of monomeric and dimeric NaPDBA, with the monomeric NaPDBA producing the ISC triplet. It would be possible to selectively make dimers or multimers by introducing functional groups to control inter-chromophore interactions or by covalently connecting chromophore units.

We have added this discussion in the revised manuscript (page 7, line 47-page 8, line 2).

(3) It doesn't seem obvious to me that lower MW power (approx. 2/3) required to maximize the DNP effect for quintets vs. triplets is inherently valuable. This claim is also undercut by the fact that the peak for DNP efficiency of the triplet is broad and appears to already be at 90% of its maximum at the peak power of the quintet.

Efficient polarization transfer requires strong microwave irradiation, and heating of samples by microwave irradiation is a major problem, especially in aqueous high-polarity systems. Therefore, reduction of microwave intensity is generally important for DNP applications. As the reviewer pointed out, the DNP efficiency of the triplet is broad, which may be related to the broad ESR spectrum of the triplet and the magnetic field sweep. As mentioned above, the most significant aspect of the present study is that the spin degrees of freedom of SF were utilized for the first time in aqueous DNP, and the microwave dependence observed here confirms the concept of using SF-derived quintets in DNP.

The revised manuscript focuses more on the importance of this proof-of-concept, and we have added the following sentences, "We form supramolecular assemblies of a few pentacene chromophores and use SF-born quintet spins to achieve DNP of water-glycerol, the most basic biological matrix, as evidenced by the dependence of nuclear polarization enhancement on magnetic field and microwave power." (page 2, line 6-9), "we show that the magnetic field dependence of nuclear polarization enhancement matches well with the ESR line shape and quintets with higher Rabi frequencies can cause DNP at lower microwave intensities than a conventional triplet ¹⁷, confirming the DNP based on the

polarized quintet spins.” (page 3, line 25-28), “This result further confirmed that the SF-derived polarized quintet state is successfully utilized for the DNP application.” (page 12, line 7-9), “The magnetic field dependence and microwave intensity dependence of DNP ensure that nuclear polarization was enhanced using quintet-derived polarization.” (page 13, line 23-25)

(4) The section about aggregation and the associated absorption spectra seems inconsistent. The absorption shift observed for the NaPDBA is assigned to aggregation but not accompanied by peak broadening or splitting. By the dimer structures given, H aggregates should be formed, resulting in a blue shift. This is not observed. Why? The CD complex shows considerable broadening, suggesting multiple intermolecular species. Much of the interpretation of aggregates is based on the MD simulations, which may not capture the true picture. NOESY is performed, but exactly how those data are used to verify the aggregate models is not clear. It seems that DOSY would also be a useful method. The authors should do a better job of describing the connection between analytical methods and the aggregate structures shown.

Fig. S1 shows the time variation of the distance between the pentacene centers of the mass (d_{COM}) and the angle (θ) between d_{COM} and the nearest distance between the pentacene units (d_{min}) obtained from MD simulations of NaPDBA in water-glycerol at 300 K and NaPDBA- γ CD in water-glycerol at 243 K. Both d_{COM} and θ fluctuate less in NaPDBA- γ CD than in NaPDBA, which is reasonable since NaPDBA dimers are encapsulated in γ CD. In NaPDBA, the distance between pentacene units is relatively far and in various orientations, which probably gave a slightly red-shifted and sharp absorption peaks. In NaPDBA- γ CD, the pentacene units associate at a closer distance which makes orbital overlap between the two chromophores. In the presence of orbital overlaps, we cannot simply classify J- or H-aggregate assuming point-dipole approximation; the excitonic coupling depends sensitively on the relative geometry between the chromophores and molecular orbitals. It has been reported that a few Å displacements affect the sign of the inter-chromophore interaction, demonstrating both J-aggregate-like and H-aggregate-like spectral changes at face-to-face geometry (Yamagata, H. et al., J. Phys. Chem. B 116, 14494-14503 (2012)). According to our MD simulation, the shift along the long-axis of the pentacene backbones varies by a few Å. The variation would result in the simultaneous presence of J-aggregate-like and H-aggregate-like dimers in the system, resulting in the broad absorption spectra of

NaPDBA- γ CD.

We have added these discussion in the revised manuscript (page 5, line 12-16; page 5, line 39-page 6, line 3).

Fig. S1. (A) Definition of distance between the pentacene centers of the mass (d_{COM}) and time dependences of d_{COM} of (B) NaPDBA in water-glycerol at 300 K and (C) NaPDBA- γ CD in water-glycerol at 243 K for the last 5 ns of MD simulations. (D) Definition of angle (θ) between d_{COM} and d_{min} which denotes the nearest distance between the pentacene units and time dependences of θ of (E) NaPDBA in water-glycerol at 300 K and (F) NaPDBA- γ CD in water-glycerol at 243 K for the last 5 ns of MD simulations. (G) Time-averaged d_{COM} and θ : circles and squares represent those values of NaPDBA in water-glycerol at 300 K and NaPDBA- γ CD in water-glycerol at 243 K, respectively; triangles represent those values of NaPDBA- γ CD in water-glycerol at 243 K using initial structure in which d_{COM} is different. (H) Time averaged of d_{COM} along the direction of the short axis (d_s) and of the long axis (d_l). The curves and plots in different colors

indicate different pentacene dimers in the system.

Following the Reviewer's suggestion, we additionally measured DOSY to further support the assembly structures of NaPDBA- γ CD and NaPDBA- β CD complexes (Fig. S9). Deuterated water was used as the solvent because the viscosity of the water-glycerol mixture was too high for reliable DOSY measurements. γ CD-derived NMR peaks were observed around 3.5-5.0 ppm and NaPDBA-derived NMR peaks around 7.5-8.5 ppm, both showing a decrease in diffusion coefficient by mixing of γ CD and NaPDBA. The diffusion coefficient of NaPDBA decreased from $2.7 \times 10^{-10} \text{ m}^2 \text{ s}^{-1}$ to $1.6 \times 10^{-10} \text{ m}^2 \text{ s}^{-1}$ and that of γ CD decreased from $2.2 \times 10^{-10} \text{ m}^2 \text{ s}^{-1}$ to $2.0 \times 10^{-10} \text{ m}^2 \text{ s}^{-1}$ upon mixing. Importantly, the diffusion coefficients of NaPDBA and γ CD were almost the same after mixing, which supports that NaPDBA and γ CD form a complex. Similarly, the mixing of NaPDBA and β CD resulted in the decrease of the diffusion coefficients of NaPDBA and β CD from $2.7 \times 10^{-10} \text{ m}^2 \text{ s}^{-1}$ to $1.8 \times 10^{-10} \text{ m}^2 \text{ s}^{-1}$ and from $2.3 \times 10^{-10} \text{ m}^2 \text{ s}^{-1}$ to $2.0 \times 10^{-10} \text{ m}^2 \text{ s}^{-1}$, respectively, confirming the formation of the NaPDBA- β CD complex.

We have included this discussion in Fig. S9.

Fig. S9. DOSY spectra of NaPDBA , CD and these mixture. (A) DOSY spectra of [NaPDBA] = 1 mM (purple), [βCD] = 2 mM (red) and a mixture of [NaPDBA] = 1 mM and [βCD] = 2 mM (blue) in D₂O at room temperature. Residual water was used as internal standard. (B) DOSY spectra of [NaPDBA] = 1 mM (purple), [γCD] = 2 mM (red) and a mixture of [NaPDBA] = 1 mM and [γCD] = 2 mM (blue) in D₂O at room temperature. Residual water was used as internal standard.

Deuterated water was used as the solvent because the viscosity of the water-glycerol mixture was too high for reliable DOSY measurements. γCD-derived NMR peaks were observed around 3.5-5.0 ppm and NaPDBA-derived NMR peaks around 7.5-8.5 ppm, both showing a decrease in diffusion coefficient by the mixing of γCD and NaPDBA. The diffusion coefficient of NaPDBA decreased from $2.7 \times 10^{-10} \text{ m}^2 \text{ s}^{-1}$ to $1.6 \times 10^{-10} \text{ m}^2 \text{ s}^{-1}$ and that of γCD decreased from $2.2 \times 10^{-10} \text{ m}^2 \text{ s}^{-1}$ to $2.0 \times 10^{-10} \text{ m}^2 \text{ s}^{-1}$ upon mixing. Importantly,

the diffusion coefficients of NaPDBA and γ CD were almost the same after mixing, which supports that NaPDBA and γ CD form a complex. Similarly, the mixing of NaPDBA and β CD resulted in the decrease of the diffusion coefficients of NaPDBA and β CD from $2.7 \times 10^{-10} \text{ m}^2 \text{ s}^{-1}$ to $1.8 \times 10^{-10} \text{ m}^2 \text{ s}^{-1}$ and from $2.3 \times 10^{-10} \text{ m}^2 \text{ s}^{-1}$ to $2.0 \times 10^{-10} \text{ m}^2 \text{ s}^{-1}$, respectively, confirming the formation of the NaPDBA- β CD complex.

(5) The pump-probe TAS section shows a straightforward production of TT, but the interpretation of the CD complex is overly exotic. It invokes three different sequentially population versions of TT. I don't understand how these are distinct electronically, especially considering that S₁ and TT should not be close to isoenergetic. I think it is more likely that inhomogeneity is playing a role, leading to different timescales and spectra for TT formation. Do the conformers suggest for the EPR section play a role in the TA dynamics also?

We appreciate the reviewer's comment. We agree with the possibility that inhomogeneity in the CD complex plays a role and leads to different rates of TT formation, resulting in apparent sequential state transfer in the global analysis. However, the distinct spectral feature shown in the TAS of NaPDBA- γ CD is also different from the TT state observed in NaPDBA: the peak positions are all different from the TT observed in NaPDBA (515 nm for NaPDBA, and 521 nm for the quick component of NaPDBA- γ CD). Also, significant peak shifts in ps-time range were also observed, which cannot be explained by simply assuming the inhomogeneity. If inhomogeneity is a dominant factor, we should see dynamic line narrowing rather than monotonical peak shift. In addition, S₁ state of dimer can be stabilized by the excitonic coupling between the tightly packed pentacenes, which may realize nearly isoenergetic condition between S₁ and TT in NaPDBA- γ CD. We admit that the dynamics would likely be affected by both inhomogeneity and fluctuation in the exciton dynamics, but attempting global analysis with three state model is reasonable to roughly capture and compare the dynamics.

To emphasize the potential importance of inhomogeneity as reviewer's comment, we have added the following sentences, "Note that significant peak shifts in ps-time range were also observed, which cannot be explained by simply assuming the inhomogeneous broadening owing to different conformers of the NaPDBA- γ CD complex. We attempted global fitting with a three-component sequential model (Fig. 3H-J). The global fitting resulted in successful fitting to the experimentally observed TAS." (page 7, line 25-29).

(6) There is never discussion of the $^3(\text{TT})$ population, despite the possibility that it could form from $^5(\text{TT})$.

It is known that $^3(\text{TT})$ can be generated from $^5(\text{TT})$ and is not directly generated from $^1(\text{TT})$, as this reviewer mentioned. However, $^5(\text{TT})$ couples with $^1(\text{TT})$ and deactivates together before $^3(\text{TT})$ is generated from $^5(\text{TT})$. Even if $^3(\text{TT})$ is generated, it would be difficult to observe because it disappears immediately by TTA. Actually, even a few microseconds after the generation of $^5(\text{TT})$, no $^3(\text{TT})$ -like ESR signal was observed, and the triplet retained its ISC-derived spectral shape (Fig. R1C, D). The ISC-derived triplet and the $^3(\text{TT})$ signal are expected to be observed at similar magnetic field, but the single-exponential decay of the triplet signal suggests that no observable amount of $^3(\text{TT})$ was generated (Fig. S23).

Fig. R1. Time-resolved ESR spectra of (A, C) NaPDBA and (B, D) NaPDBA- γ CD in water-glycerol (1:1) at 143 K ($[\text{NaPDBA}] = 1 \text{ mM}$, $[\gamma\text{CD}] = 5 \text{ mM}$) at 1 μs (A, B) and 3 μs (C, D) after photoexcitation at 527 nm and simulated spectra.

Table S3 shows calculated matrix elements of the absolute magnitudes of the spin Hamiltonian of $\text{abs}(\mathbf{H}_{\text{TTB}})$ based upon the TT geometry ($\alpha = 20$ degrees, $\beta = 0$ degrees, $\gamma = 20$ degrees, $\theta_B = 14$ degrees, $\phi = 0$ degrees) in Fig. S31, as treated in *J. Phys. Chem. B* **2020**, *124* (42), 9411-9419. While the interaction between the ^5TT and ^1TT is large for generating $^5\text{TT}_{\pm 2}$ and $^5\text{TT}_0$ with the coupling values of 3.5×10^9 rad/s, the interactions between ^3TT and ^1TT is zero. Also, the interactions between ^5TT and ^3TT minor being ca. 10^8 rad/s. Thus, the generations of the ^3TT states are neglected in the present multiexciton. It is thus concluded that the ^3TT is not created from the quintet multiexciton in the presence of the strong exchange coupling.

Fig. S31. Angles defined for geometry of the triplet pair for the ZFS principal axes represented by X1, Y1, Z1 for triplet 1 and by X2, Y2, Z2 for triplet 2 in the TT multiexciton. (A) Polar angles, (θ , ϕ) and (B) Euler angles, (α , β , γ).

Table S3. Matrix elements of the magnitudes (10^{11} rad/s) of the spin Hamiltonian of $[\text{abs}(\mathbf{H}_{\text{TTB}})]$ calculated for the TT_B state with the B_0 direction of $(\theta_B, \phi = (90^\circ, 45^\circ))$ of the external magnetic field in the (X1, Y1, Z1) coordinate in Figure R1 in the presence of the exchange coupling of $J_B = -0.8$ T.

	${}^5\text{TT}_{+2}$	${}^5\text{TT}_{+1}$	${}^5\text{TT}_0$	${}^5\text{TT}_{-1}$	${}^5\text{TT}_{-2}$	${}^3\text{TT}_{+1}$	${}^3\text{TT}_0$	${}^3\text{TT}_{-1}$	${}^1\text{TT}$
${}^5\text{TT}_{+2}$	3.9231	0.0004	0.0001	0.0000	0.0000	0.0010	0.0007	0.0000	0.0345
${}^5\text{TT}_{+1}$	0.0004	3.3931	0.0002	0.0001	0.0000	0.0016	0.0007	0.0005	0.0019
${}^5\text{TT}_0$	0.0001	0.0002	2.8395	0.0002	0.0001	0.0012	0.0000	0.0012	0.0343
${}^5\text{TT}_{-1}$	0.0000	0.0001	0.0002	2.2637	0.0004	0.0005	0.0007	0.0015	0.0018
${}^5\text{TT}_{-2}$	0.0000	0.0000	0.0001	0.0004	1.6676	0.0000	0.0007	0.0010	0.0379
${}^3\text{TT}_{+1}$	0.0010	0.0016	0.0012	0.0005	0.0000	2.2371	0.0003	0.0001	0.0000
${}^3\text{TT}_0$	0.0007	0.0007	0.0000	0.0007	0.0007	0.0003	2.8485	0.0003	0.0000
${}^3\text{TT}_{-1}$	0.0000	0.0005	0.0012	0.0015	0.0010	0.0001	0.0003	3.3665	0.0000
${}^1\text{TT}$	0.0345	0.0019	0.0343	0.0018	0.0379	0.0000	0.0000	0.0000	5.6349

These discussions have been included in the revised manuscript (page 9, line 19-27), Table S3).

(7) This is probably understood by those working on DNP, but why is the magnetic field in Fig 1C modulated?

In order to transfer the spin polarization from triplet electron spins to nuclear spins, it is necessary to match the Larmor frequency of nuclear spins in a static magnetic field with the Rabi frequency of electron spins under microwave irradiation. However, since the ESR linewidth of triplet is broaden due to the hyperfine interactions and different orientations, it is impossible to resonate all triplet electron spins at once in a certain magnetic field. The magnetic field sweep allows all electron spin packets in the swept range to participate in the polarization transfer, increasing the efficiency of DNP.

We have added an explanation of the magnetic field sweep to the main text as follows, “Due to the broadening of ESR linewidth by hyperfine coupling and different molecular orientations, only a small fraction of spin packets can satisfy the Hartmann-Hahn condition at a time. One of the solutions of this problem is to use the ISE sequence that sweeps the external magnetic field during the microwave irradiation, so that more spin packets can be used for the polarization transfer.” (page 10, line 19-page 11, line 4).

(8) What is the difference between the Euler angles and dipolar angles in Table S2? The caption refers to reference 57 in the main text, but there are only 42 references.

We appreciate this comment. We have corrected “dipolar angles” to “polar angles”. The polar angles represent the position of the second NaPDBA relative to the first NaPDBA. In this case, $\theta = 35^\circ$, $\varphi = 0^\circ$ means that the NaPDBAs are slightly displaced from each other in the long axis direction. The Euler angle represents how much the molecular plane of the second NaPDBA is tilted with respect to the molecular plane of the first NaPDBA. For example, $\alpha = 0^\circ$, $\beta = 0^\circ$, $\gamma = 0^\circ$ means the orientations of NaPDBAs are parallel, and $\alpha = 90^\circ$, $\beta = 10^\circ$, $\gamma = -90^\circ$ means the second NaPDBA is oriented with a 20 degree tilt in the direction of the long axis relative to the first NaPDBA.

This information has been included in the revised manuscript (Fig. S31).

Fig. S31. Angles defined for geometry of the triplet pair for the ZFS principal axes represented by X1, Y1, Z1 for triplet 1 and by X2, Y2, Z2 for triplet 2 in the TT multiexciton. (A) Polar angles, (θ , φ) and (B) Euler angles, (α , β , γ).

(9) Pg 9, lines 7-8—it should be said explicitly that the reason the orientation for which populations were calculated (field aligned with the molecular plane) would dominate because of the statistics of powder samples.

To illustrate the molecular orientation involved in the EPR transition, we made a mapping of the transverse magnetization at the field strength of “X, Y” in Fig. S21A from the spin polarized EPR spectrum of ^5TT state. This is performed by distributing the transverse magnetizations (intensities of the EPR spectra) at 326.4 mT to all possible directions of the external magnetic fields. From this imaging map, the microwave absorption signal at this field position is dominated by the ESR transitions for the magnetic fields directing to the X1-Y1 plane in the triplet pairs. On the other hand, the mapping of the transverse magnetization at another field strength of 310.4 mT showed that the ESR transition is contributed by the out-of-plane direction (Z1) (Fig. S21B). Overall, the absorptive microwave transitions of $^5(\text{TT})_0 \rightarrow ^5(\text{TT})_{+1}$ at 326.4 mT is dominantly contributed by the in-plane field directions, even though the powder sample was employed in the present DNP experiment.

We have added the following text regarding these discussions, “The largest ESR signal was observed when the magnetic field (B_0) is parallel to the pentacene backbone (Fig. S21).” (page 9, line 7-8).

Fig. S21. (A) Mapping of the transverse magnetization at the field strength of “X, Y” in Fig. S22A from the spin polarized EPR spectrum of ^5TT state. (B) Mapping of the magnetizations for the field strength of “Z” in Fig. S22. The mappings of the magnetization were performed from the computations of electron spin polarization¹ for all possible directions of the external magnetic field as reported previously^{2,3}.

(10) Labeling of the two different dimers as TT1 and TT2 is confusing, as numerical subscripts and superscripts are used to denote spin multiplicity elsewhere. I would suggest TTA and TTB instead. I think there may also be a related mistake in labeling on

pg. 10, line 13

We are thankful to this kind suggestion. We have changed the labeling from TT_1 and TT_2 to TT_A and TT_B in the revised manuscript.

(11) Abstract says “sole way” to generate quintet, text says “sole way without heavy metals”

We appreciate this comment. We have changed the word “sole way” to “effective way” in the revised manuscript.

(12) Pg 2, line 27 – DNP for medical applications isn't typically grouped with the “second quantum revolution”

We agree with the Reviewer and have removed the description of second quantum revolution.

(13) Pg 2, line 28 – More detailed theoretical models (e.g. JDE model) indicate that true selective population of 5TT_0 requires shared molecular axes and their alignment with B_0

Related to the comment (9), it is true that aligning the orientation of molecules with respect to the magnetic field, in addition to aligning the intermolecular orientation, enables the preferential population of 5TT_0 , and the ESR peak becomes very sharp, which is advantageous from a DNP perspective. However, as mentioned in our response to the comment (1), it is possible to select the electron spins to be used for DNP by adjusting the magnetic field. Selective population can also be obtained regardless of orientation by forming a multimer and using mixing in a state which the two triplets are weakly coupled to each other with J values smaller than D and E values.

(14) Pg 3, line 5— it is stated that up to 200% quintet yield (where triplet is probably intended) can be achieved, but the reference cited does not even mention quintets

As the reviewer pointed out, the 200% is the description about the triplet, and we've removed it to avoid any misunderstanding.

(15) Pg 3, line 19—Again using language that suggests selective population of $^5\text{TT}_0$, when models and TR-EPR spectra in the paper suggest that's not the case for an unoriented sample

We have changed the description to describe it as preferential population.

(16) “Because the ESR spectral analysis showed a similar singlet recombination rate constant from the $^1(\text{TT})$ dimer, the lower enhancement factor of the NaPDBA- γ CD complex was probably due to the triplet deactivation¹⁶ caused by triplet-triplet annihilation (TTA) between neighboring pentacenes, as observed in ns-17 TAS.”

How was singlet recombination rate obtained from EPR? Is $^1(\text{TT})$ supposed to read TT_1 ?

We'd appreciate this comment. The singlet recombination constant k_{REC} represents the rate of singlet TT dimer $^1(\text{TT})$ deactivation to the ground state. $^1(\text{TT})$ and $^5(\text{TT})$ are assumed to be in equilibrium. The k_{REC} value was estimated based on change in EPR signal intensity from immediately after photoexcitation to 2-3 μs later. The change of EPR signal intensity was well explained with k_{REC} of $1.0 \times 10^6 \text{ s}^{-1}$ for NaPDBA and $2.0 \times 10^6 \text{ s}^{-1}$ for NaPDBA- γ CD. We found these k_{REC} values explained the experimental results better than those used in the original manuscript and thus changed them in the revised manuscript. $^1(\text{TT})$ is not TT_1 (or TT_A/TT_B). It is singlet TT dimer ($^1(\text{TT}_A)$ and $^1(\text{TT}_B)$).

This explanation has been included in Table S2.

Replies to the comments of Reviewer 2:

This is a very interesting paper showing that singlet fission processes, which generate a non-equilibrium spin-2 quintet state, can be used to enhance nuclear polarization by DNP procedures. The experimental results are modest, showing NMR signal enhancements of around 20, but do provide intriguing evidence of a new family of DNP agents and procedures that could well be very useful in certain circumstances. Certainly the concept is novel and constitutes an impressive piece of science crossing over from photovoltaics to NMR, MRI, and eventual possible clinical applications.

The paper is, for the most part, very well-written. However there are few points of concern which, in my view, should be addressed.

We are very pleased that this reviewer has so highly evaluated our work. Below are our responses to the individual comments.

* In the title, the abstract, and in many places in the text, the experiment is called “biological”. It isn’t. The method may hold some promise for eventual biological applications, but this is most definitely not “biological DNP”. I have to call this what it is: the gratuitous, and ill-advised, use of buzzwords, to enhance the chance of publication. There is nothing biological about the described experiments.

We appreciate this comment. As the reviewer pointed out, our concept is expected to contribute to biological applications in the future, but the current experiments were not conducted on biological samples. We have revised the manuscript so as not to use the word biological too easily. We removed the word “biological” from the title and the new title is “Singlet fission as a polarized spin generator for dynamic nuclear polarization”.

* on page 2, line 31, it is stated that the singlet fission “preserves its total spin”. This is manifest nonsense, since the singlet fission process goes from $S=0$ to $S=2$. The authors appear to mean that the magnetic quantum number m_S is preserved, at least approximately. That may well be true under certain circumstances, but the phrase should be reworded so that the science is accurate, and it should be indicated within which regime or set of approximations the m -value is preserved. Certainly this is not an

absolute selection rule, which the authors appear to imply.

We appreciate the reviewer for this suggestion to improve the accuracy of the description about magnetic quantum number. We have revised the text as follows, “It has been revealed that the $|^5(\text{TT})_0\rangle$ spin sublevel with a magnetic quantum number $m_s = 0$ can be preferentially populated among five quintet sublevels because the state immediately after photoexcitation is a spinless singlet state, and the transition with preserving total spin magnetic quantum number m_s preferentially occurs during the generation process of spin-correlated triplet pairs¹³⁻¹⁶.” (page 2, line 29-34).

* on page 3, line 21, the “Hartmann-Hahn condition” is indicated. Whatever is going on in figure 1B and 1C, it is certainly not Hartmann-Hahn matching. In general, the description of the procedure for the transfer of electron spin-polarization to nuclear polarization is very poor and poorly referenced. This aspect of the paper needs to be greatly improved, in my opinion.

We appreciate this comment. To improve the description of the polarization transfer procedure, we have added the following sentences, “The integrated solid effect (ISE) sequence was used for the DNP experiments (see Fig. 1B, C, S24).^{28,35} The sequence starts from laser irradiation to produce the polarized electron spins. Then, microwave and magnetic field sweep are applied simultaneously. By matching the Rabi frequency of electron spins and the Larmor frequency of nuclear spins (proton in this experiment) under the microwave irradiation, the electron spin polarization can be transferred to nuclear spins. This is called as Hartmann-Hahn condition ($\gamma_e B_1 = \gamma_H B_0$), where γ_e, γ_H is the gyromagnetic ratio of the electron and proton spin, respectively. B_0 is the external magnetic field, and B_1 is the oscillating magnetic field by microwave irradiation. Note that B_1 is applied perpendicular to B_0 , and proportional to the square root of microwave power. Due to the broadening of ESR linewidth by hyperfine coupling and different molecular orientations, only a small fraction of spin packets can satisfy the Hartmann-Hahn condition at a time. One of the solutions of this problem is to use the ISE sequence that sweeps the external magnetic field during the microwave irradiation, so that more spin packets can be used for the polarization transfer. The transferred polarization is automatically diffused throughout the sample by dipolar interaction between proton

spins. By repeating the above sequence, the proton polarization builds up while balancing with the spin-lattice relaxation.” (page 10, line 10- page 11, line 7).

In addition, we have modified Fig. 1B to better explain the polarization transfer process and made an additional figure to illustrate the Hartmann-Hahn condition (Fig. S24).

Fig. 1. Schematic illustration of DNP using SF-born quintet electron polarization. (A) Nuclear spins in the thermal equilibrium state. (B) Polarization transfer from electron spins in the quintet state generated by photo-induced SF to nuclear spins and the subsequent diffusion of hyperpolarized nuclear spins. (C) Pulse sequence of quintet/triplet-DNP. (D) Molecular structures of NaPDBA and γ -cyclodextrin (γ CD) and supramolecular assembly of only NaPDBA and the NaPDBA- γ CD inclusion complex.

(E) Absorption spectra of NaPDBA in water-glycerol at 143 K (black), NaPDBA- γ CD in water-glycerol (1:1) at 143 K (blue), and NaPDBA in methanol at room temperature (red). The concentrations of NaPDBA and γ CD were 1 mM and 5 mM, respectively.

Fig. S24. Schematic illustration of the Hartmann-Hahn matching condition. The nuclear spins precess with the Larmor frequency ω_H around the external magnetic field B_0 in the laboratory frame. The electron spins precess around the effective magnetic field $B_{eff} = \sqrt{B_1^2 + \Delta B^2}$ with the frequency $\omega_e = \gamma_e B_{eff}$ in the rotating frame, where B_1 is the power of the irradiated microwave perpendicular to the external magnetic field, ΔB is the offset between ω_{e0} and ω_{MW} ($\Delta B = (\omega_{e0} - \omega_{MW})/\gamma_e$). When the Hartmann–Hahn matching condition $\omega_H = \omega_{eff}$ is satisfied, the polarization transfer occurred most efficiently ^{4,5}.

*on page 10, line 2, we read that “the magnetic field is aligned with the ESR peaks”. What is this supposed to mean? Needs rewording to be made scientific, clear, and rigorous.

We appreciate this comment. Following the reviewer’s suggestion, we have revised the description as follows, “The DNP sequence shown in Fig. 1C was performed with a central magnetic field of 629 mT, the field corresponding to the quintet ESR peak. The sweep range of magnetic field was about 10 mT (Fig. S25). We note that different magnetic fields were used for ESR and DNP. ESR was measured at a resonance frequency of 9.0 GHz to compare with common X-band data, while DNP was measured at 17.3 GHz, the frequency at which microwave amplifiers are available as high as

possible to increase the T_1 of ^1H .” (page 11, line 11-16).

* in the caption to Fig.5, the “ISE sequence” is mentioned. As far as I can tell this is unexplained.

In general, the description of the electron-to-nuclear polarization transfer procedure, and its operation, are very poor and need considerable improvement.

As mentioned above, we have added the description about the electron-to-nuclear polarization transfer and the ISE sequence in page 10, line 10- page 11, line 7.

Replies to the comments of Reviewer 3:

The authors prepare pentacene derivative (NaPDBA) supramolecular assemblies using beta or gamma cyclodextrin in water or water-glycerol matrices and report the transient EPR and transient absorption spectra of these materials. The prepared NaPDBA complexes are found to favor NaPDBA monomers or multimers (including dimers) as is evidenced by presence of singlet fission (SF) in some of these materials which would require a close spatial proximity between two of the NaPDBA molecules and the absence of SF in the monomer cases. The optical excitation of these materials produce both a polarized triplet (via ISC) and polarized quintet state (via ISC/SF) and the authors demonstrate that this optically generated spin polarization from both the triplet and quintet states may be used to enhance ^1H NMR spectra via ISE DNP. To my knowledge, this work would be the first demonstration of polarization transfer between a polarized quintet electron ensemble and ^1H via ISE DNP. The original study is well done and thorough, with experimental results supported by molecular dynamics simulations. This study would be of interest to a broad audience interested in material science, magnetic resonance and fundamental spin physics. The work is also of significance to the fields of hyper polarized NMR and quantum information science. I would recommend that this paper be published after revisions and following the authors' answers to the following comments and questions:

It is our great pleasure that this reviewer has found the significance and broad interest of our work. We reply to each comment below.

General Comments and Questions:

1. The statement that singlet fission is the sole way to generate a spin quintet state in organic materials is not true. Please see the following publications in which quintet ground states (Q_0) are formed through photolysis in ketocarbenes <https://pubs.acs.org/doi/pdf/10.1021/ja00353a055> and non-Kekulé structures <https://pubs.acs.org/doi/pdf/10.1021/ja00212a006>. Please address this by revising your statements throughout the text. Quintets (Q_1) in organic materials are also formed in triplet-triplet annihilation upconversion, albeit they are less stable. Perhaps the authors meant to say that long lived Q_1 only accessible via SF? Or polarized Q_1 ? Please clarify this point.

We appreciate this thoughtful suggestion. We wanted to say that SF is useful to create a polarized quintet state. We have changed the word “sole” to “effective”, and revised the text as “Singlet fission (SF), converting a singlet excited state into a spin-correlated triplet-pair state, is an effective way to generate a spin quintet state in organic materials.” (page 2, line 1-2) and “SF provides the effective method to create spin polarized quintet states in organic molecular systems without heavy metals.” (page 2, line 26-28).

2. Not all experimental details were included. What was the sweep range (in mT) used for the magnetic field for polarization transfer for the triplet DNP? What about for the quintet DNP? The authors only report on the sweep rate, not the range. The range should be reported for all of the DNP experiments.

The relationship between the NMR resonance frequency offset and the applied voltage yielded a slope of -0.2386 mT/V. Since a sweep voltage of 50 V was applied in the ISE sequence, this corresponds to a sweep range of about 10 mT.

We have added this information in the revised manuscript (page 16, line 21-25, Fig. S25).

Fig. S25. Offsets from the NMR resonance frequency at 27.98 MHz when various voltages are applied to the field sweep circuit. Deionized water was used for NMR measurement. A 50 V sweep in the ISE sequence in this study corresponds to a 10 mT

sweep.

3. The authors report that the lower enhancement factor of the NaPDBA gamma CD complex is likely due to triplet deactivation, however, a simpler explanation would be that the polarization lifetime of the NaPDBA-gammaCD complex is nearly 3 times shorter than that of the NaPDBA aggregates, wouldn't this be the main contribution to the decreased enhancement? Especially given that the length of each DNP cycle is longer than 10 microseconds?

As this reviewer pointed out, the factor that determines the magnitude of sensitization by DNP involves not only the intensity of the ESR signal, but also its lifetime. We consider that the ESR intensity mainly affects the enhancement factor, as the signal-to-noise ratios of the ESR spectra of NaPDBA and NaPDBA- γ CD are very different, but certainly differences in lifetime may also have an effect.

We have added this discussion in the revised manuscript (page 11, line 37-43).

4. The authors mention that an enhancement at the quintet position was not observed for the NaPDBA-betaCD complex. Could the authors include this data in figure 5? As this would be the most convincing evidence that the polarization is derived from the quintet.

We appreciate this kind suggestion. We agree with the reviewer that it is very important to show the convincing evidence in the figure of the main text. We have included the magnetic field dependence of the ^1H NMR signal enhancement by DNP in Figure 5. With NaPDBA and NaPDBA- γ CD, signal enhancement of 30% and 44% was observed at the magnetic field (629 mT) corresponding to the quintet ESR peak compared to the magnetic field (641 mT for NaPDBA and 639 mT for NaPDBA- γ CD) corresponding to the triplet ESR peak, respectively (Fig. 5C, D). On the other hand, with NaPDBA- β CD, this ratio was remarkably low (9.9%), supporting the absence of quintet-induced DNP (Fig. 5E).

We have included this discussion in the revised manuscript (page 11, line 20-31).

Fig. 5. DNP using SF-born quintet electron spin polarization. (A), (B) ^1H -NMR signals under thermal conditions (black lines, 5 scans every 10 min) and after quintet-DNP (red lines, ISE sequence for 5 min, 1 scan) of water-glycerol (glycerol- d_8 : D_2O : H_2O = 5:4:1) containing (A) NaPDBA and (B) NaPDBA- γ CD at 100 K ([NaPDBA] = 1 mM, [γ CD] = 5 mM). The photo-excitation wavelength and frequency were 527 nm and 500 Hz, respectively. DNP was performed by matching the magnetic field to the quintet peaks. The microwave power and frequency were 20 W and 17.30 GHz, respectively, the laser powers were 2.7 W for (A) and 1.5 W for (B). The magnetic field sweep width was 10 μs . Magnetic field dependence of the signal intensity of the ^1H NMR by DNP and time-resolved ESR spectra in water-glycerol containing (C) NaPDBA, (D) NaPDBA- γ CD, and (E) NaPDBA- β CD ([NaPDBA] = 1 mM, [β CD] = [γ CD] = 5 mM). Water-glycerol glass (glycerol- d_8 : D_2O : H_2O = 5:4:1) was used for the DNP measurement at 100 K. ISE sequence for 20 s (C, E) and 30 s (D) and 1 scan; microwave power and frequency were 20 W and 17.25 GHz, respectively; laser power: 1.5 W; magnetic field sweep width: 10 μs . Water-

glycerol glass (glycerol:H₂O = 5:5) was used for the time-resolved ESR measurements at 143 K. ESR spectra were integrated for 10 μ s after photoexcitation in order to compare the DNP profile with the ISE sequence for 10 μ s. (F) Microwave power dependence of DNP enhancement. Triplet-DNP was performed at 27.4 MHz (ISE sequence for 10 s and 4 scans with a laser power of 2.7 W, microwave frequency of 17.30 GHz and sweep width of 25 μ s). Quintet-DNP was performed at 26.9 MHz (ISE sequence for 10 s and 10 scans with a laser power of 2.7 W, microwave frequency of 17.30 GHz and sweep width of 10 μ s).

5. Unless the sweep range for the ISE (mT) is very narrow about the quintet absorption peak < 10 mT, wouldn't the authors expect some polarization contribution from the triplet? Please clarify this point. Did the authors also measure the DNP experiment using the emissive quintet peak to observe an inverted 1H signal? It would be interesting to see the full DNP profile (if the authors have the data) and see how well it matches with the trEPR spectrum.

As mentioned above, the sweep field range used in this study was ~10 mT, which was narrow enough to distinguish between triplet-DNP and quintet-DNP. As mentioned in the response to the previous comment #4, when NaPDBA- β CD was used, only 9.9% of the signal enhancement was observed at the magnetic field of the quintet ESR peak (629 mT) compared to that of the triplet ESR peak (641 mT). There is a slight contribution of the triplet due to its broad ESR spectrum, but it is sufficient to distinguish between DNP using the quintet or the triplet.

Following the Reviewer's suggestion, we additionally measured the full range of DNP profile (Fig. 5C-E). Since ISE sequence was carried out for 10 μ s, the ESR spectrum integrated for 10 μ s after photoexcitation was compared to the DNP profile. The emissive enhanced NMR peaks were observed by using emissive quintet ESR peak for NaPDBA and NaPDBA- γ CD (Fig. 5C, D). Meanwhile, no emissive NMR peak was observed for NaPDBA- β CD at the magnetic field corresponding to the emissive quintet ESR peak (Fig. 5E). There was an approximate correlation between ESR and DNP profiles, confirming that DNP was performed using triplets and quintets, respectively.

We have included this discussion in the revised manuscript (page 11, line 20-31).

6. Figure S11. The authors should discuss the complexation energies obtained through

MD rather than just showing a region on a snapshot of the MD, as these calculated values would be more informative to support the observed trEPR and TA data.

We appreciate your important remark. For quantitatively evaluating the stability of the complex in the simulated system, a potential of mean force (PMF) was analyzed: the higher the value of the PMF of the complex, the more stable the complex. The PMF for pulling away one of γ CD forming the complex with NaPDBA were calculated for the quantitative comparison of the stability of the complex in water-glycerol between NaPDBA- γ CD at 300 K and 243 K. According to the previous study, the PMF for each system was obtained from a series of umbrella sampling (US) simulations, where the energy minima of their umbrella potentials were located at equal intervals along the direction of pulling γ CD and the calculated probability density distributions were overlapped. For preparing the initial positions of two series of the US simulations, nonequilibrium steered MD simulations were performed using the structure after the 20 ns equilibration runs. The center of the mass of one molecule of γ CD molecules forming the complex was pulled away in one direction using an umbrella potential. The values of the PMF were calculated from each set of the US simulations by the weighted histogram method (WHAM).

The PMFs for pulling away one γ CD molecule of the 2:2 inclusion complex of NaPDBA- γ CD in water-glycerol until it unfolded the NaPDBA dimer were 7.9 ± 2.1 and 21.1 ± 2.4 kJ/mol at 300 K and 243 K, respectively (Fig. S15). Therefore, it shows that the inclusion complex of NaPDBA- γ CD at 243 K was more stable than that at 300 K. Since the thermal energy of the complex consisting of 4 molecules at 300 K can be estimated about 10 kJ/mol, the complex of NaPDBA- γ CD in water-glycerol at 300 K should not be energetically stable. These results are consistent with the experimental results that the inclusion complexes of NaPDBA- γ CD were unstable at room temperature in water-glycerol.

The discussion mentioned above was added to the revised manuscript and Supplementary Information (page 6, line 5-13, Fig. S15). The models and method of additional MD simulations were also written in the section of the methods of the main text.

Fig. S15. PMF profiles of 2:2 inclusion complex of NaPDBA- γ CD in water-glycerol at 300 K (A) and at 243 K (C). Each inset snapshot illustrates the top view of the NaPDBA- γ CD complex at the moment of its collapse. Apparent probability density of 2:2 inclusion complex of NaPDBA- γ CD in water-glycerol at 300 K (B) and at 243 K (D). The curves drawn in different colors were calculated from each US simulation.

7. Figure S13. What was the excitation energy and why was a different excitation energy (rather than 600 nm) used for this dataset?

The excitation wavelength was 520 nm and intensity was kept to less than 0.1 mJ/cm². Since ns-TAS is a complimentary measurement to TR-ESR about excited states kinetics in ns- μ s range, we decided to measure the TAS using a similar excitation wavelength with TR-ESR (532 nm). We have added the information of the excitation wavelength as “(excitation: 520 nm)” in the caption of Fig. S17 (Fig. S13 in original manuscript).

8. Figure S15. Why is no triplet observed in this figure for NaPDBA-betaCD? This does not match S14.

Both figures show fs-TAS of NaPDBA- β CD, but the experimental conditions were different: Figure S18(Figure S14 in original manuscript) and S19(Figure S15 in original manuscript) were taken under water solution at room temperature, and water-glycerol solution under 143 K, respectively. Since NaPDBA shows more significant aggregation in the water solution and incomplete encapsulation of β CD, we also detect SF of a fraction of aggregated NaPDBA. However, we'd like to emphasize that the second component of EAS in the Figure S18E is mainly S₁-like spectral feature, indicating that most of NaPDBA was encapsulated by β CD.

To avoid the confusion, we added to the caption of Fig. S18 (Fig. S14 in original manuscript):

... at room temperature,

...indicated by the almost S₁-like spectral feature of EAS₂.

Formatting and grammar corrections:

1. The order in which the supplementary figures are introduced in the main text is not in order, this makes it time consuming to navigate between the main text and supplementary. Please place figure labels in order as they are introduced.

We have changed the order in which the figures are displayed as suggested by this reviewer.

2. Figures 3A and 3F would be more informative if the figure is shown from $x = 450$ to 550 nm as the features discussed in the caption cannot be seen clearly as it is currently shown.

We thank the reviewer's comment. We modified the range of x-axis of corresponding figures to 450-550 nm for better clarity of spectral evolution as shown below.

Fig. 3. fs-TAS measurements of the supramolecular assemblies. Overview of fs-TAS analysis of (A-E) NaPDBA and (F-J) NaPDBA- γ CD in water-glycerol (1:1) at 143 K ($[\text{NaPDBA}] = 1 \text{ mM}$, $[\gamma\text{CD}] = 5 \text{ mM}$). (A, F) Pseudo-2D plots of experimentally observed fs-TAS (excitation: 635 nm for NaPDBA and 600 nm for NaPDBA- γ CD), (B, G) spectral evolution of the TAS, and (C, H) temporal change of transient absorption at selected wavelengths and fitting curves from global analysis. (D, I) Evolution-associated spectra and (E, J) corresponding concentration kinetics obtained from global analysis based on sequential models.

3. Figure 4: Please show the quintet manifold total fit (as you have for the triplet - dotted line) as it is difficult to see the quality of the fit as it currently is.

We have added the sum of the ESR signals of the quintet to Figure 4 as suggested by

this reviewer. (Fig. 4)

Fig. 4. Time-resolved ESR measurements of the supramolecular assemblies. Time-resolved ESR spectra of (A) NaPDDBA and (B) NaPDDBA- γ CD in water-glycerol (1:1) at 143 K ([NaPDDBA] = 1 mM, [γ CD] = 5 mM) just after photoexcitation at 527 nm and simulated spectrum. (C) for NaPDDBA and (D) for NaPDDBA- γ CD, attributing transitions between each energy level of the quintet in the ESR spectra. The fitting parameters of the ISC-born triplet and SF-born quintet are summarized in Tables S1 and S2, respectively.

4. Typo page 10 line 8, chemical

5. Typo page 10 line 36, glassy

6. Typo page 14 line 12, were

We have corrected the typo as per this reviewer's suggestion.

REVIEWER COMMENTS

Reviewer #1 (Remarks to the Author):

The authors have changed the language around the primary claims, which is helpful. They have performed additional experiments to support their conclusions. However, there continues to be misleading language and referencing. I have made some specific comments below.

General comments:

(1) The Weiss et al. reference uses the term 'selective', but closer inspection suggests it does not mean that 5TT0 is solely populated according to their data. This is an older paper that does not contain some of the nuance discovered in more recent work. Although the TR-EPR signal may be dominated by the 5TT0 contribution under some conditions, other quintet populations can exist in significant fractions. Table S2 seems to also suggest this. Also, the Weiss et al. model requires coexistence of 5TT0 with dissociated triplets, which would necessarily lower the proportion of 5TT0 existing in the sample. Although the authors quote 100% as the theoretical limit, I continue to wonder what is the actual best overall 5TT0 polarization that can be achieved given realistic sample conditions.

(2) It has not been acknowledged in the main text that the overall (unnormalized) polarization from the ISC triplet was more effective than that of the quintet, and even normalized (Fig 5) the quintet polarization is only (slightly) higher between 15-20 W. Considering the difference in efficiency, their data does not show that better DNP can be caused at lower MW power with quintets, just that the (much smaller) effect for quintets is saturated at lower MW powers. The reason for the lower enhancement for quintets vs. what appears from the peak amplitudes in TR-EPR may be due to lifetime effects – the triplet lives much longer than quintet and therefore may be more effective at DNP.

(3) The responses to my concerns about the TA kinetic analysis were not satisfying. This addition to the manuscript is meaningless: 'The global fitting resulted in a successful fitting to the experimentally observed TAS.' Ascribing an evolution associated spectrum to a combination of species means that the kinetic model associated with the EAS is inappropriate. Some language remains unclear: 'Heavily mixed electronic states' – what does it mean? Electronic coherence typically dephases in <1 ps at room temperature. Is that consistent with the kinetics observed for the decay of the assigned mixed state to the pure state?

Reviewer #2 (Remarks to the Author):

The authors have addressed my comments well in the revised version. I now find the article suitable for publication in Nature Communications.

As a minor change, I suggest that references are given to the NOVEL experiment, when discussing the transfer of polarization between electrons and nuclei by establishing Hartmann-Hahn match between the electron nutation frequency in the microwave field and the nuclear Larmor frequency in the static field. Suitable references are: [doi.org/10.1016/0022-2364\(88\)90190-4](https://doi.org/10.1016/0022-2364(88)90190-4) and doi.org/10.1080/00268970801998262

Reviewer #3 (Remarks to the Author):

After thoroughly reviewing the revised manuscript I have seen that the authors have addressed my comments and concerns and have made great improvements to the figures and analysis. I would suggest that the now revised work be published.

Replies to the comments of Reviewer 1:

The authors have changed the language around the primary claims, which is helpful. They have performed additional experiments to support their conclusions. However, there continues to be misleading language and referencing. I have made some specific comments below.

We are glad that this Reviewer found the revisions we made to the manuscript to be helpful. We also appreciate the opportunity to further improve the quality of the manuscript by correcting the misleading language and referencing this Reviewer pointed out. Below are our responses to each comment.

General comments:

(1) The Weiss et al. reference uses the term 'selective', but closer inspection suggests it does not mean that $^5\text{TT}_0$ is solely populated according to their data. This is an older paper that does not contain some of the nuance discovered in more recent work. Although the TR-EPR signal may be dominated by the $^5\text{TT}_0$ contribution under some conditions, other quintet populations can exist in significant fractions. Table S2 seems to also suggest this. Also, the Weiss et al. model requires coexistence of $^5\text{TT}_0$ with dissociated triplets, which would necessarily lower the proportion of $^5\text{TT}_0$ existing in the sample. Although the authors quote 100% as the theoretical limit, I continue to wonder what is the actual best overall $^5\text{TT}_0$ polarization that can be achieved given realistic sample conditions.

As the Reviewer pointed out, in the Weiss et al. reference (Weiss, L. R. et al., *Nat. Phys.* **13**, 176-181 (2016)), $^5(\text{TT})_0$ was selectively generated among the five quintet TT pairs, but a dissociated triplet was also generated at the same time. The Weiss et al. paper deals with crystals, so the formation of dissociated triplet is inevitable, while our dimer structure does not generate such a dissociated triplet. In fact, our ESR spectra did not show the characteristic A/E/A/E spin polarization pattern of the triplet dissociated from $^5(\text{TT})_0$ (Matsuda, S. et al., *Chem. Sci.*, **11**, 2934-2942 (2020)), suggesting that the contribution of the dissociated triplet is almost negligible. This is consistent with the DLS and MD simulation results that NaPDBA forms dimers in water-glycerol and does not form larger aggregates. Therefore, we have removed discussion based on the Weiss et al. reference because it may be misleading to discuss the selective $^5(\text{TT})_0$ population

in the dimer structure based on the Weiss et al. paper, following the Reviewer's suggestion. The present novel dimer preparation is thus promising for the actual best overall $^5(\text{TT})_0$ polarization in the realistic sample conditions (although it would be desirable to reduce the ISC-triplet polarization by further structural control), as described below.

A rigorous *JDE* model for selective $^5(\text{TT})_0$ generation in dimer structures has been reported (Smyser, K. E., Eaves, J. D., *Sci. Rep.*, **10**, 18480 (2020)). According to this *JDE* model, $^5(\text{TT})_0$ is generated as a nearly pure quantum state by making the exchange interaction between chromophores sufficiently large and by making the principal axes of the two chromophores parallel to each other and to the Zeeman field. The validity of this model has been well confirmed by the fact that it explains well the experimental results of oriented crystalline samples (Lubert-Perquel, D. et al., *Nature Commun.*, **9**, 4222 (2018); Rugg, B. K. et al., *Proc. Natl. Acad. Sci. U.S.A.*, **119**, e2201879119 (2022)). Notably, in our dimer system, the exchange interaction between chromophores is sufficiently large in the order of GHz, and the principal axes between chromophores are close to parallel, satisfying some of the requirements of the *JDE* model. We agree with the reviewer that other quintet sublevels such as $^5(\text{TT})_2$ and $^5(\text{TT})_{-2}$ were also populated, agreeing with the *JDE* model. In the previous revision, we discussed that the ESR transition used for triplet-DNP has almost no contribution of $^5(\text{TT})_2$ and $^5(\text{TT})_{-2}$ and explained this point in the revised main text (page 10, line 5-19). The remaining challenge is to orient the chromophores with respect to the magnetic field, which would be achieved by using an orienting agent (Krebs, P., et al., *J. Magn. Reson.* **22**, 359–373 (1976)) or orienting dispersed nanocrystals (Kaneko, Y., et al., *J. Mater. Chem.* **15**, 253–255 (2005)).

We have revised the manuscript accordingly with appropriate references (page 2, line 29-35; page 3, line 8-9; page 10, line 27-30; page 14, line 30-32; ref. 19-21, 42, 44, 45).

(2) It has not been acknowledged in the main text that the overall (unnormalized) polarization from the ISC triplet was more effective than that of the quintet, and even normalized (Fig 5) the quintet polarization is only (slightly) higher between 15-20 W. Considering the difference in efficiency, their data does not show that better DNP can be caused at lower MW power with quintets, just that the (much smaller) effect for quintets is saturated at lower MW powers. The reason for the lower enhancement for quintets vs. what appears from the peak amplitudes in TR-EPR may be due to lifetime effects – the triplet lives much longer than quintet and therefore may be more effective at DNP.

The main focus of this study is to demonstrate that polarized quintet electron spins

produced by singlet fission can be utilized in DNP. The DNP with quintet is supported by the fact that DNP efficiency is maximized at lower microwave intensities than with triplet. Although triplet gave the larger DNP effect than quintet in the present system, it is possible to suppress the formation of ISC-derived triplet by selectively synthesizing dimers, and it would be possible to improve the quintet DNP performance more by selectively generating $^5(\text{TT})_0$ among the quintet sublevels by controlling the orientation of the dimers relative to the magnetic field as mentioned in our response to the comment (1). The lifetime of the quintet was long enough to conduct DNP, so it could be an excellent DNP polarization source if selective $^5(\text{TT})_0$ generation is achieved. We have revised the manuscript accordingly (page 12, line 31-34).

(3) The responses to my concerns about the TA kinetic analysis were not satisfying. This addition to the manuscript is meaningless: 'The global fitting resulted in a successful fitting to the experimentally observed TAS.' Ascribing an evolution associated spectrum to a combination of species means that the kinetic model associated with the EAS is inappropriate. Some language remains unclear: 'Heavily mixed electronic states' – what does it mean? Electronic coherence typically dephases in <1 ps at room temperature. Is that consistent with the kinetics observed for the decay of the assigned mixed state to the pure state?

We thank the reviewer for the clarification. We thoroughly revised the description of the TAS and the results of the global fitting results and carefully redescribed the languages with respect to the discussion of S_1 -TT mixing as follows. We experimentally observed the signal around 510-520 nm emerged rapidly after photoexcitation concurrently with the broad absorption around 450-500 nm. This indicated prompt generation of the excited states containing $^1(\text{TT})$ states character owing to the stronger electronic coupling between the chromophores. The spectral shape of the initial TA was also different from that observed for the bare NaPDBA aggregates. These observations can be explained by the model of either (1) initial generation of the S_1 -TT mixed adiabatic electronic states as suggested in the chromophores with strong electronic interaction (e.g., discussed in *Acc. Chem. Res.* 2020, 53, 1957–1968; *Acc. Chem. Res.* 2013, 46, 6, 1321–1329; *J. Phys. Chem. B* 2021, 125, 6945–6954), or (2) simultaneous detection of coherent and incoherent SF, which could result from the dynamic fluctuation and inhomogeneity of the system. Since the first EAS components clearly show different spectral shapes, which contain the characteristic spectral shape of S_1 and $^1(\text{TT})$, we conclude that the strong electronic

coupling between the adjacent pentacene moieties leads to the mixed adiabatic state of S_1 and $^1(\text{TT})$ or rapid SF within the time resolution of our system (~ 100 fs). We decided to denote the initial component as “[S_1 -TT]” to emphasize the indistinguishably mixed character of the state indicated by the spectral shape. Consequently, the transition to the hot $^1(\text{TT})$ state occurs in 0.79 ± 0.01 ps via SF. Because the timescale is likely quicker than the reorganization of the relative geometry of the paired chromophores (e.g., excimer formation), we denote the second and third components as TT* and TT^{rel}, respectively. To distinguish the above mechanisms of the SF in the NaPDBA- γ CD system, more sophisticated spectroscopy, such as coherent two-dimensional electronic spectroscopy, will be needed [*J. Am. Chem. Soc.* 2018, 140, 51, 17907–17914]. We admit that this would certainly be an exciting direction, but here we judged that these are beyond our focus in the current manuscript. Nevertheless, we emphasize that our TAS successfully confirmed the acceleration of the generation of $^1(\text{TT})$ by more than a factor of three compared to the bare NaPDBA system.

We have revised the manuscript accordingly with appropriate references (page 7, line 22-page 8, line 9).

Replies to the comments of Reviewer 2:

The authors have addressed my comments well in the revised version. I now find the article suitable for publication in Nature Communications.

As a minor change, I suggest that references are given to the NOVEL experiment, when discussing the transfer of polarization between electrons and nuclei by establishing Hartmann-Hahn match between the electron nutation frequency in the microwave field and the nuclear Larmor frequency in the static field. Suitable references are: [doi.org/10.1016/0022-2364\(88\)90190-4](https://doi.org/10.1016/0022-2364(88)90190-4) and doi.org/10.1080/00268970801998262

We are pleased that this reviewer considers our paper is now suitable for publication in Nature Communications. We have cited the suggested references where we illustrate the Hartman-Hahn matching condition (ref. 4, 6 in SI).

Replies to the comments of Reviewer 3:

After thoroughly reviewing the revised manuscript I have seen that the authors have addressed my comments and concerns and have made great improvements to the figures and analysis. I would suggest that the now revised work be published.

It is our great pleasure that this reviewer has found our manuscript sufficiently improved to merit publication.

REVIEWERS' COMMENTS

Reviewer #1 (Remarks to the Author):

I thank the reviewers for considering my requests seriously. I am now satisfied that the prior issues are resolved.